# Mapping Overlaps in Benchmarks through Perplexity in the Wild

**Siyang Wu**[†]    **Honglin Bao**[†‡]    **Sida Li**[†]    **Ari Holtzman**[*‡]    **James Evans**[*‡]

Data Science Institute, University of Chicago

[†] The first three authors, S.W., H.B., and S.L., co-led the project and contributed equally

[*] The last two authors, A.H. and J.E., co-supervised the project

[‡] Correspondence to: {honglinbao, aholtzman, jevans}@uchicago.edu

## Abstract

We introduce benchmark signatures to characterize the capacity demands of LLM benchmarks and their overlaps. Signatures are sets of salient tokens from *in-the-wild* corpora whose model token perplexity, reflecting training exposure, predicts benchmark performance. We extract them via stepwise forward selection with linear regression in a meta-evaluation spanning 32 LLMs and 89 benchmarks across diverse domains. We then analyze how these signatures relate to both the semantic similarity of benchmark questions and the correlation structure of model performance. While performance correlations are uniformly high and semantic overlaps stay in a narrow mid-range, benchmark signatures reveal more nuanced structure. For instance, they uncover substantial overlap between benchmarks in knowledge and reasoning tasks, whereas benchmarks in culture- and humanity-oriented domains show low similarity with each other. Unlike raw performance correlations, which are influenced by benchmark-*orthogonal* factors such as question formats, signatures are robust to such confounds. We further identify cross-functional overlaps between logic, math, language, instruction following, and cultural/world modeling, with coding emerging as the most isolated function, interacting only moderately with the ability of detecting missing information. Qualitative analysis shows that only the knowledge signature aligns with actual knowledge, suggesting that LLM semantic organization may differ from human conceptual structure. Together, these findings offer insights into benchmark validity, LLM sensitivities, and the landscape of interconnected LLM capacities. We have open-sourced the code and data in this GitHub repository.

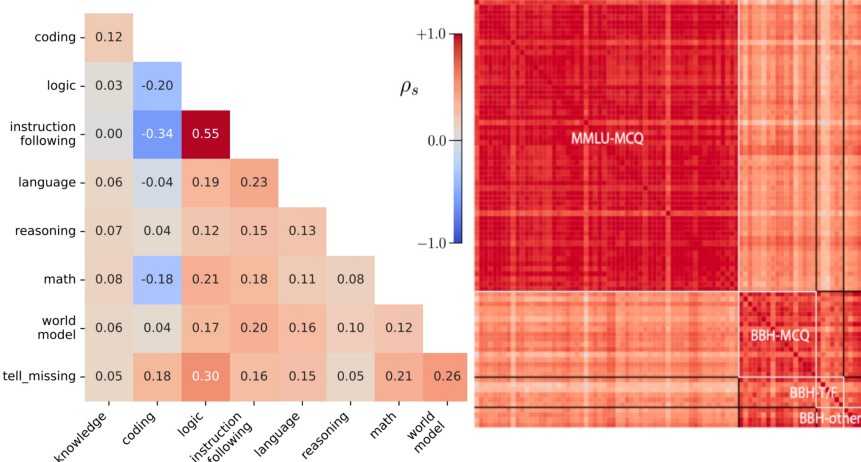

Figure 1: **Left**: Signature correlations across functions. **Right**: Performance alignments are biased (red areas: benchmark families or question formats: Multi-Choices vs. True-False).

# 1 INTRODUCTION

Benchmarks have been central in the growth of large language models (LLMs): they catalyze progress, standardize evaluation, and enable systematic cross-model comparisons, thereby influencing the trajectory of AI research. The community has witnessed an accelerating proliferation of benchmarks across a wide range of LLM abilities, such as reasoning (Tafjord et al., 2020) and agentic capabilities (Zhu et al., 2025), as well as real-world scenarios such as finance (Zhang et al., 2023) and safety (Mou et al., 2024). The dedicated "Datasets and Benchmarks" track in leading venues such as NeurIPS and KDD highlights both the importance and steady growth of this area. Each year witnesses many new benchmark papers. From 252 submissions to the NeurIPS Datasets and Benchmarks Track in 2021 to 1,820 in 2024 [1], the number of benchmark papers has increased more than sevenfold. While these resources often claim to assess distinct capabilities, it is frequently unclear whether they truly do so, or whether they merely capture narrow proxies, prompt-specific heuristics, or even overlapping skills that have already been extensively tested elsewhere, making them less unique and useful than advertised. This raises critical questions: *Do we really need such a vast and ever-expanding suite of benchmarks? How much overlap exists across them?* Answering this question will also reveal the converse: *What areas of capability are sparsely underrepresented by benchmarks and might benefit from more?*

In this paper, we undertake a comprehensive meta-evaluation with a particular focus on identifying and analyzing benchmark overlap, which we define as the degree to which two benchmarks evaluate a shared set of model capabilities. To capture overlap in a principled way, we examine it from three complementary perspectives. At the **semantic** level, we assess whether the questions in two benchmarks substantially overlap in content or intent; if so, their redundancy is intrinsic. At the **performance** level, the mainstream level in benchmark agreement studies (Perlitz et al., 2024), we test whether models show highly correlated performance across two benchmarks, indicating that they measure related underlying abilities even if under surface semantic differences. Finally, at the **benchmark signature** level - introduced by us in Section 3 - we move beyond tasks and outcomes to characterize the distributional fingerprint of benchmarks, defined by token-level perplexity patterns on large-scale in-the-wild corpora.

Why do in-the-wild corpora effectively encode benchmark characteristics? The abilities measured by benchmarks - commonsense, factual memory, scientific reasoning, programming skills, and more - do not emerge out of thin air. They stem from the diverse real-world text patterns encountered by the model. In-the-wild corpora, consisting of large-scale, naturally authored, multi-domain text and code (news, forums, encyclopedias, textbooks and notes, papers, documentation, blogs, and repositories), are produced for human communication rather than adapted for benchmark design. They are rich in task-bearing structure (question–answer, problem–solution, claim–evidence, instruction–execution), redundancy (the same function expressed in many ways), and breadth. This breadth of distribution - likely unique to in-the-wild data - forms the "soil" from which such capabilities grow, and also the source from which benchmark questions are drawn. Even if a benchmark item never appears verbatim, its "function" recurs pervasively: unit-aware arithmetic in recipes ("double 1½ cups"), commonsense causality in narratives ("the glass shattered after being dropped"), claim → measurement → inference chains in scientific abstracts, code repair patterns in GitHub issues ("off-by-one in loop; fix bounds"), and even schema–query mappings ("customers with orders in last 30 days"). Focusing only on synthetic or benchmark-adjacent data risks capturing artifacts of test design. In-the-wild data, by contrast, mirrors the true distribution that gives rise to these abilities, making the overlap between capacity exposure and benchmark competence not accidental but expected.

Perplexity provides a useful lens for quantifying relationships between skill exposure and benchmark performance. Low perplexity on a passage suggests that the model has seen similar linguistic and conceptual patterns during training and is familiar with the content. High perplexity, by contrast, indicates unfamiliarity and underrepresentation. Thus, the distribution of perplexity values across large corpora serves as a fingerprint of the model's training exposure and more or less acquired capacity. Importantly, because different benchmarks stress different capabilities, they map onto different perplexity distributions when probing across the same corpus. In other words, corpora encode benchmark signatures because benchmarks are not foreign entities imposed on the model after train-

---

[1]https://papercopilot.com/

ing, but rather structured samplings of capabilities that themselves emerge from the distribution of in-the-wild data. Perplexity serves as the bridge between exposure and benchmark performance, making it possible to identify and characterize these signatures without requiring direct evaluation on the benchmark itself [2]. We therefore leverage perplexity as the basis and covariate for salient token selection and signature formation [3]. The following three levels in this work provide a holistic framework: semantics address task design, performance captures model behavior, and signatures reveal a fingerprint of model capacity. The overlap between benchmarks across each of these levels highlights the interconnected capacity space - an oft-discussed yet difficult-to-formalize concept and so represents a promising tool for evaluating benchmark validity. This rationale is illustrated in Figure 2.

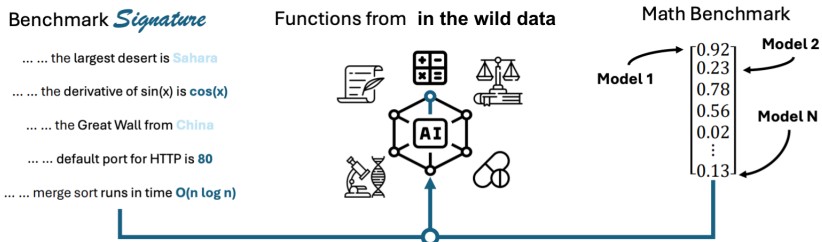

Figure 2: Overview of the rationale of how in-the-wild corpora implicitly encode the benchmark signature, knowledge exposure (capacities), as well as benchmark performance.

---

**Definition: Benchmark Signature**

A *benchmark signature* is defined as a set of salient tokens $T$, extracted from large-scale in-the-wild corpora, such that the perplexity of a collection of language models $M$ on $T$ is highly predictive of their performance on the benchmark.

---

To achieve the overall process, we make the following three contributions:

- We introduce a systematic framework for measuring benchmark relations and especially their overlap across three levels: **semantic**, **performance**, and **signature** derived from model perplexity.

- We develop a forward selection and regression-based pipeline to extract these signatures by mining and filtering token-level perplexity statistics from *in-the-wild* corpora.

- We uncover unexpected overlaps between widely used benchmarks. While these benchmarks are intended to test a specific ability - such as logic - and their problem sets do align with human intuitions about logic, in practice they often measure instruction-following ability in language models instead. This reveals the potential issue of benchmark design and actual execution, as well as the interconnected space of LLM capabilities.

## 2 SEMANTIC OVERLAPS AND PERFORMANCE OVERLAPS

**General Notation.** We denote the collection of $m$ LLMs by $M_1, \ldots, M_m$ and the set of $n$ benchmarks by $B_1, \ldots, B_n$. For any quantity defined jointly over a model-benchmark pair—such as a performance metric $y$ - we write $y_{i,j}$ to indicate the metric value corresponding to model $M_i$ evaluated on benchmark $B_j$. Unless otherwise specified, all vectors are column vectors and are set in bold lowercase, e.g. $\mathbf{x} \in \mathbb{R}^d$. Matrices are represented with capital letters in bold, e.g. $\mathbf{X} \in \mathbb{R}^{n \times m}$.

**Semantic-Level Overlap.** For benchmarks $B_a, B_b$ with question text sets $Q_a, Q_b$, let $n_{\min} = \min\{|Q_a|, |Q_b|\}$. Let $k$ be the embedding dimension, $f : \text{text} \to \mathbb{R}^k$ be a sentence transformer (e.g. MPNet encoder; Song et al. (2020)), and let $s(x, y)$ be the cosine similarity between $x, y \in \mathbb{R}^k$.

---

[2]More discussions about prior works see A.1.

[3]More details see Section 3 and Appendix A.2.

Because benchmarks vary in size (i.e., number of questions), which could bias results, we estimate overlap via size-matched bootstrapping similarity: for $T = 1000$ times, we draw $\{q\}_t^{(a)} \subset Q_a$ and $\{q\}_t^{(b)} \subset Q_b$ independently with $|\{q\}_t^{(a)}| = |\{q\}_t^{(b)}| = n_{\min}$, encode each item with $f$. Also, $Concate()$ means concatenating a list of texts into a single string within each set, and computing cosine similarity:

$$\widehat{A}_{\text{sem}}(B_a, B_b) = \frac{1}{T}\sum_{t=1}^{T} s\big(f(concat(\{q\}_t^{(a)}), f(concat(\{q\}_t^{(b)}))\big)$$

Overall, this mitigates sample-size bias and yields a more robust similarity estimate. Full procedural details appear in Appendix A.4.1. The overlap between $B_a$ and $B_b$ is defined by the $\widehat{A}_{\text{sem}}(B_a, B_b)$.

**Performance-Level Overlap.** For each benchmark $B_a$, let $\mathbf{y}_{:,a} \in \mathbb{R}^m$ be the vector of model performances on $B_a$ (one entry per model). The performance-level overlap between two benchmarks $B_a$ and $B_b$ is the Spearman rank correlation between their model-marginalized performance vectors:

$$\rho(B_a, B_b) = \text{corr}\big(\text{rank}(\mathbf{y}_{:,a}), \text{rank}(\mathbf{y}_{:,b})\big)$$

Thus, the overlap between $B_a$ and $B_b$ under performance-level is defined by the spearman correlation which is $\rho(B_a, B_b)$.

## 3 MINING BENCHMARK SIGNATURES FROM IN-THE-WILD DATA

---

**Algorithm 1** Obtaining signature for benchmark $B_j$

---

**Input:** Data "in the wild" $\mathcal{D}$, Benchmark $B_j$, a list of LLMs $M_1, ..., M_m$
**Output:** Signature $S_j$
1: $y_{:,j} \leftarrow B_j$ with $M_1, ..., M_m$ ▷ Generate performance column vector on benchmark $j$.
2: $\mathcal{T} \leftarrow \mathcal{D}$ ▷ Processing in-the-wild data into tokens with preceding context, specifically, the prefix consisting of the 30 preceding segments, where each segment corresponds to a segment defined by space.
3: $\mathbf{P} \leftarrow \mathcal{T}$ With $M_1, ..., M_m$ ▷ Generate the token-level perplexity covariate matrix.
4: $\mathcal{T}'_j \leftarrow \text{AIC}(\text{THRUSHPREFILTER}(\mathbf{P} \sim y_{:,j}))$ ▷ Perform Thrush pre-filtering first; then stepwise AIC feature selection on the covariate matrix $\mathbf{P}$ against the performance vector $y_{:,j}$ of benchmark $B_j$ to obtain salient tokens.
5: Retrieve $S_j$ from mapping $\mathcal{T}'_j$ in $\mathbf{P}$
6: **return** $S_j$

---

The overall process of mining signatures can be found in Algorithm 1 and its details can be found in Appendix 4. Let $d$ denote the number of "in-the-wild" tokens (in our case, tokens drawn from large-scale pretraining corpora[4]), denoted as $\mathcal{T} = \{t_1, \ldots, t_d\}$, where $d$ typically scales to billions. For any benchmark $B_j$, our objective in extracting its *benchmark signature* is to isolate a subset of salient tokens $\mathcal{T}'_j = \{t'_1, \ldots, t'_{d'}\} \subset \mathcal{T}$ that are maximally informative in explaining variations in LLM performance. We formalize this as a regression problem: let $\hat{\mathbf{y}}_j := (y_{1,j}, \ldots, y_{m,j})^\top \in \mathbb{R}^m$ denote the performance of $m$ language models on benchmark $B_j$. The covariate matrix $\mathbf{P} \in \mathbb{R}^{m \times d}$ contains token-level perplexities, where entry $\mathbf{P}_{ij} \equiv p_{ij}$ corresponds to the perplexity of token $t_j$ under LLM $M_i$. The challenge lies in the high-dimensional regime ($d \gg m$, where $d \approx 8.45 \times 10^9$ and $m = 32$), where classical regression approaches are ill-posed. To make progress, we must uncover and exploit latent structural properties of the problem. In particular, we put forward a key assumption and a follow-up question:

1. **Sparsity:** Most token-level perplexities are uninformative for predicting benchmark performance, with only a small fraction carrying predictive signals. **[Assumption 1]**

2. **Extraction:** Assuming sparsity holds, what methods can effectively identify and extract these predictive signals from the overwhelming background of noise? **[Question 1]**

---

[4]Progressing from fine to coarse granularity, we have token-, chunk-, and document-level perplexities. We provide more experimental results of why the token-level operation is the best. Details are shown in Appendix A.2.

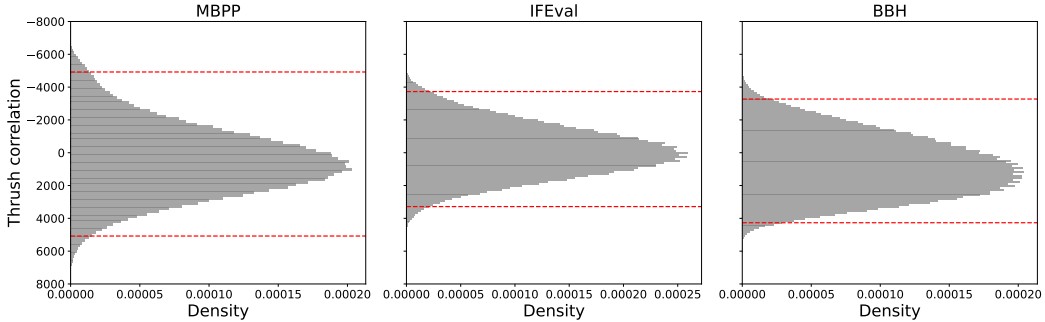

Figure 3: Distribution of Thrush correlations in pre-selection phases; red vertical lines mark the 1st and 99th percentiles, highlighting that few features are highly correlated with performance.

Together, they motivate our below regression-based framework for mining benchmark signatures, which leverages high-dimensional inference techniques to disentangle signal from noise and recover benchmark-specific fingerprints of token-level perplexity.

### 3.1 TOKEN-LEVEL FILTERING WITH PERPLEXITY CORRELATIONS

Answering **[Q1]** by fitting a full multivariate regression model is computationally intractable given that the number of tokens $(d)$ is orders of magnitude larger than the number of models $(m)$. We therefore adopt a pragmatic and efficient two-stage approach, beginning with a screening step to drastically reduce the feature space. Specifically, for each benchmark, we perform a token-by-token *correlation screening*. We compute a *robust correlation coefficient* between each token's perplexity vector and the benchmark performance vector. This screening is highly efficient, requiring linear time in the number of features, $O(md)$, and allows us to observe the empirical distribution of these coefficients. In a sparse regime, we expect this distribution to be sharply peaked at zero, with small tails representing potentially informative tokens.

A known limitation of this screening approach is its reliance on *marginal, univariate* correlations. It evaluates each token in isolation, potentially overlooking features that are predictive only in a multivariate context (e.g., suppressor tokens that explain residual variance). However, we argue this approach is theoretically and empirically well-justified in our specific problem setting for the following reasons:

1. **Justification from the Ultra-High Dimensional Regime:** Our problem, with $d \gg m$, resides in the ultra-high dimensional setting. Theoretical frameworks developed for this regime, such as Sure Independence Screening (SIS; Fan & Lv (2008)), provide formal guarantees for marginal screening. The "sure screening property" ensures that, under sparsity and certain regularity conditions, correlation-based filtering can discard the vast majority of irrelevant features while retaining the true predictive signals with very high probability. We further explain how several key conditions of SIS are plausible in our context in Appendix A.3.

2. **Empirical Precedent in Data Selection:** This screening methodology has demonstrated strong empirical success in the related domain of data selection for training. Prior work has successfully used document-level perplexity correlations to filter large corpora, improving downstream model performance (Thrush et al., 2025; Shum et al., 2025). Their success provides compelling evidence for the practical utility of correlation screening as a robust heuristic for identifying informative signals in LLM-related data.

While there exist various methods for robust correlation calculation, there is no single "silver bullet"; the choice is often guided by the specific properties of the data. In particular, we highlight the two robust correlation coefficients introduced in the aforementioned data selection literatures.

**Definition 3.1** (Thrush Correlation (Thrush et al., 2025))**.** Fixing the $j$-th token $t_j \in \mathcal{T}$. Let $\mathrm{rank}_j(p)$ denotes the rank of $p$ among $\{p_{1,j}, \cdots, p_{m,j}\}$ and $\mathrm{sign}(\cdot)$ be the sign function, we denote

$$\gamma_j = \sum_{1 \le k < l \le m} \mathrm{sign}(y_{k,j} - y_{l,j})(\mathrm{rank}_j(p_{k,j}) - \mathrm{rank}_j(p_{l,j})) \tag{1}$$

as the Thrush correlation coefficient. This coefficient is a variant of Kendall's $\tau$ (Kendall, 1938), measuring the concordance between model performance and perplexity ranks. It counts the number of model pairs where the model with better performance also has a lower perplexity rank (a concordant pair), and subtracts the number of pairs where this is not the case (a discordant pair), making it robust to the absolute magnitude of perplexity values.

**Definition 3.2** (Pre-select Correlation (Shum et al., 2025))**.** Letting $Z = \frac{m(m-1)}{2}$ to be a normalizing factor, and $(1), ..., (m)$ be the sorted indices by LLM performances (i.e. $y_{(1),j} \le y_{(2),j} \le \cdots \le y_{(m),j}$), the Pre-select correlation coefficient is defined as:

$$\eta_j = \sum_{1 \le k < l \le m} \mathbf{1}\{p_{k,j} > p_{l,j}\}/Z \tag{2}$$

The Pre-select coefficient computes the fraction of model pairs that are "misordered" by their token perplexities relative to their benchmark performance. In an ideal scenario where lower perplexity perfectly predicts higher performance, this sum would be zero; a value of 0.5 would indicate a random, uninformative relationship.

Once these robust correlation coefficients are calculated for all $d$ tokens, we employ a simple quantile-based threshold to screen the feature space, retaining approximately the top 1% of tokens with the strongest signal. Figure 3 presents the empirical distributions of the Thrush coefficients for three representative benchmarks. In all cases, the distributions are sharply peaked around a central value (indicating a random relationship), with thin tails representing tokens that are highly correlated with performance. This characteristic shape provides compelling empirical support for our sparsity hypothesis (**[Q1]**): the vast majority of token perplexities are uninformative, while a small, identifiable subset carries a significant predictive signal.

### 3.2 Refining signatures with forward selection regression

The correlation screening successfully isolates a candidate set of potentially informative tokens, satisfying our goal of drastically reducing the search space. However, this filtering alone is insufficient to define a robust benchmark signature for two primary reasons. First, the filtered set is likely to contain redundant features; for instance, several top-ranked tokens might represent the same underlying linguistic phenomenon and thus offer overlapping predictive information. Second, a true signature should not only identify important tokens but also capture their *conditional* importance – their predictive power given the other tokens already in the model.

To address these challenges and distill a final, parsimonious signature, we employ a second-stage multivariate variable selection procedure. Our general framework can accommodate various high-dimensional regression techniques suited for the $d' > m$ regime (where $d'$ is the number of filtered tokens, $d' \approx 1.69 \times 10^7$), including penalized methods like Lasso (Tibshirani, 1996), Ridge (Hoerl & Kennard, 1970), or Elastic Net (Zou & Hastie, 2005). In practice, we opt for a greedy forward selection approach, which we find builds interpretable and effective models. This method iteratively constructs the signature by adding the single token from the candidate pool that yields the greatest improvement to the model's fit, penalized by its added complexity.

To guide this selection process, we use the Akaike Information Criterion (AIC; Bozdogan (1987)), which provides a principled trade-off between explanatory power and model size, mitigating the risk of overfitting. The process terminates when no additional token can improve the model's AIC score by a meaningful amount. The complete two-stage process – combining the initial correlation screening to create a candidate set with the subsequent forward selection to derive the final signature – is formalized in Algorithm 4.

### 3.3 Signature-Level Overlap

Consider two benchmark signature vectors, $S_1$ and $S_2$, each including several pieces of context (30 pieces separated by space) + the salient token. We use 32 models to process these signatures, reading

their respective pre-contexts, producing the last token-level perplexities and calculating overlaps. If the models are confused to a similar degree by both signatures, that is a strong indicator that the two benchmarks align. Since some "weak" models consistently produce high perplexity, we normalize each model's perplexity into its z-score within the model. We then compute the mean of z-scored perplexities of the two benchmark signatures within each model and the Spearman correlation between these two mean lists to represent the signature-level overlap, aligning with performance level results and indicating models' relative relation of perplexity and skill familiarity on the signature. Refer to Appendix A.5.1 for a formalized walk-through.

We further examine the robustness of our framework in four dimensions. First, **the robustness of design**: we examine the generalizability of the framework, specifically, whether the regression merely overfits the observed data rather than generalizing to unseen models, and the extent to which base abilities tested across benchmarks influence the results. Second, **the robustness of methods**: we assess the robustness of the regularization and screening methods used in the paper and compare them to their alternatives. Third, **the robustness of parameters**: we study the robustness of parameter choices such as the 1% pre-filtering threshold. Fourth, **the robustness of data**: how to approximate the "in-the-wild" corpora and whether it impacts the major conclusion. We found that our framework is robust across all dimensions, and notably, it is easily replicable on a smaller scale with limited computational resources. These details can be found in A.7 (robustness) and A.8 (computational cost).

## 4 RESULTS

Our experiments are conducted on 32 models and 89 benchmarks, including many of the most widely used ones. We extract benchmark signatures from the open dataset *RedPajama* (Weber et al., 2024). See Appendix A.5 for the full details of the experimental setup.

### 4.1 SIGNATURES CAN BETTER DISTINGUISH BENCHMARKS THAN SEMANTICS AND MODEL PERFORMANCE

We first examine how the overlap distribution looks across three levels, as illustrated in Figure 4. To minimize inductive bias, we assign broader categories to these benchmarks using the official labels from MMLU (Hendrycks et al., 2021), Big-Bench Hard (Suzgun et al., 2022), ifeval benchmark (Zhou et al., 2023), and MBPP (Austin et al., 2021). In signature overlap (panel a), on the left, we compare within-category overlap against the average cross-category overlap. To reduce the impact of benchmark category size, we ensure each category pair is weighted equally. We then use the mean of cross-category overlaps to represent the overall cross-category overlap and apply this consistently throughout the paper. We observe that overlap is higher within certain categories such as reasoning, science, and social science knowledge, which is expected: benchmarks designed around the same high-level intent tend to align, whereas pairs such as chemistry vs. history benchmarks overlap far less. Within the humanities and world models, overlaps are generally lower than those in cross-category comparisons. A closer look at these benchmarks suggests that the lower similarities stem from their emphasis on diverse cultural contexts - for example, world-model evaluations that assess understanding of culture-specific phenomena like movies and sports - and their reliance on processing humanities-based material such as history from a wide range of countries and regions. Furthermore, within a category, certain benchmarks align more strongly than others. This forms a dense "red clique," identified by extracting the maximum clique from the overlap graph. We highlight these highly aligned benchmarks on the right side. For panel (b) semantic overlap and panel (c) performance overlap, in contrast, these analyses show much weaker discriminative ability. Semantic overlap scores remain in a narrow range (typically 0.1–0.4) regardless of whether benchmarks come from the same or different categories. Conversely, performance-level overlap is almost universally high, suggesting that model performance and the semantic meaning of questions are less sensitive to category boundaries and obscure finer-grained, underlying associations between benchmarks.

At the semantic level, text embedding models such as MPNet capture surface-level similarity in how humans perceive benchmark questions (Morris et al., 2023). These representations are highly dependent on the specific descriptive intention behind a question, however, meaning the overlap remains superficial and does not reflect the underlying abilities being evaluated. In other words, identical questions do indicate overlapping benchmarks, but *different questions do not necessarily*

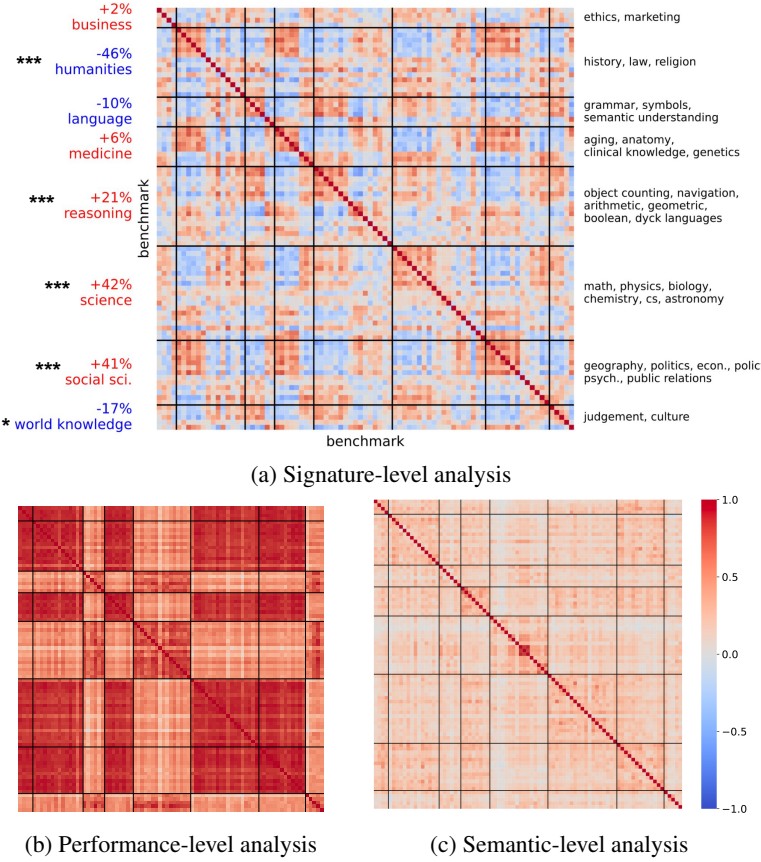

(a) Signature-level analysis

(b) Performance-level analysis      (c) Semantic-level analysis

Figure 4: Three levels of benchmark relation analysis. The signature-level analysis demonstrates substantially stronger discriminative ability compared to both semantic- and performance-level analyses. All heatmaps are presented using a consistent color range from -1 to 1, and panels b and c share the same row and column indices articulated in panel a. Statistical details can be found in Appendix A.6. * $p < 0.05$; ** $p < 0.01$; *** $p < 0.001$.

*indicate non-overlapping ones in terms of underlying ability.* At the performance level, while some overlap was initially observed, it quickly became clear that this too fails to meaningfully separate categories. In fact, performance-level results show strong segregation: model behaviors on certain cross-category benchmarks are as closely aligned as they are within categories (evident in several segregated red areas not on the diagonal). When we examine these unexpectedly high alignments, we find that they occur within the same broad benchmark families (e.g., MMLU or BigBench-Hard) or under the same question format (e.g., True/False versus multiple-choice questions). This benchmark-orthogonal effect is even stronger than within-category overlaps - that is, MMLU history aligns more closely with MMLU chemistry than with another history benchmark. This underscores the limitations of relying on performance alone and highlights deep issues in current benchmark agreement tests. Several factors could explain this pattern: the bias may stem from post-training fine-tuning, and it could also reflect contamination of the training data, where exposure to one evaluation within a benchmark family increases the likelihood of exposure to others, thereby inflating performance correlations. Another explanation lies in model capabilities: when a model is tested for a single ability, the evaluation inevitably involves a combination of multiple common skills - at a minimum reading, instruction following, and comprehension, among others. This overlap makes behavioral alignment a less distinguishable measure.

## 4.2 THE EVALUATION BIAS IS RESOLVED BY THE SIGNATURE

Grouping the result in Figure 4 panel b, we observe that the red areas are concentrated within the same benchmark family and question format as shown in Figure 1 right panel, where two red areas

are exactly two benchmark families or question formats. We calculated pairwise correlations between benchmarks both within and across families and question formats. Since each family or format contains a highly diverse set of benchmarks - essentially covering everything - we would expect within-family/format overlaps to be quite low, showing little difference from cross-family/format overlaps. Consistent with this expectation, the signature-level analysis reveals statistically insignificant tiny differences based on the Mann–Whitney U test, yielding results around 0. This aligns with intuition, as the signature provides a good approximation of the true overlap and variation. In contrast, the performance-level analysis shows a large value of overlap (around 0.8) and a statistically significant increase in within-family/format overlap. Our results show deep issues in current benchmark agreement tests that LLM performance may be more related to surface-level aspects of benchmarks, such as question format, suggesting both that generalization and knowledge-propagation in LLMs are limited and that current evaluation may be underestimating peak performance because of conflation of performance and competence. Using linear regression to obtain signature filters out the noise associated with the error term while preserving the underlying systematic relationships among benchmarks and performances.

### 4.3 SIGNATURES INFORM BENCHMARK DESIGN AND LLM CAPACITY SPACE

As shown in Figure 1, we compare overlaps across design functions. Several patterns emerge. First, we observe significant overlaps that align with intuition. For example, math and logic correlate at 0.21, which is close to the average within-function overlap of 0.285 and far above the average cross-function overlap of 0.105. This makes sense: solving a math problem often requires logical reasoning, and vice versa. More broadly, logic, instruction following, language, math, and world modeling (largely cultural benchmarks) form a cluster of interconnected abilities. Coding appears far less entangled with other

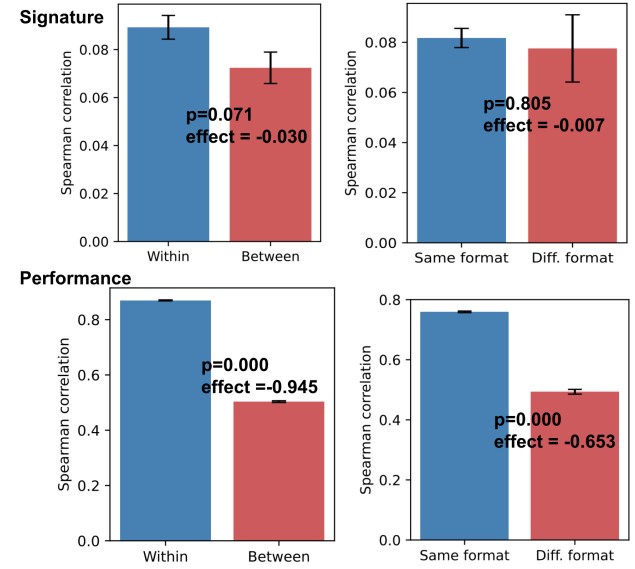

Figure 5: Biases (within/between families; same/diff. formats) are well addressed by the signature.

functions. Its low cross-function overlap suggests that coding benchmarks are comparatively "clean," in the sense that success relies more specifically on coding competence and less on auxiliary abilities. It only moderately interacts with the ability to detect missing information in a sequence. This distinctiveness might arise because coding requires highly specialized pretraining corpora such as GitHub, which is also one of the three major domains in AbsenceBench (Fu et al., 2025).

There are two broad perspectives for interpreting these results. If we *optimistically* assume that benchmarks faithfully measure what they claim, then the observed overlaps reveal a genuine interdependence of cognitive abilities. In this view, benchmarks are not "leaky," but rather reflect the multifaceted nature of capacity like math and logic. From this perspective, overlap is not noise, but evidence of underlying LLM and human capacity entanglement - the interconnected capacity space - an often-discussed but previously difficult-to-formalize concept. Alternatively, the overlaps may expose a misalignment between what benchmarks intend to measure and what they actually capture. This interpretation suggests that benchmarks are "leaky" in undesirable ways, inadvertently testing skills outside their stated domain. For example, even if math and logic are highly related, their overlap should theoretically remain lower than within-math or within-logic overlap. Yet, Figure 1 shows

cases where cross-function overlap exceeds within-function overlap - for instance, between instruction following and logic. This could imply that either within-function overlap is underestimated (due to poorly aligned benchmark design and execution (Liao et al., 2021)) or that cross-function contamination is stronger than anticipated, undermining the clarity of what each benchmark is supposed to isolate.

## 5 QUALITATIVE INTERPRETATION OF BENCHMARK SIGNATURES

What exactly are signatures? We performed a qualitative analysis of the textual signatures. Our approach uses a simple metric of textual similarity: we compare the intended function of a benchmark (for example, assessing social-science knowledge) with the textual content of its signature using the model from (Song et al., 2020). We find that when a benchmark targets knowledge in a specific field, its signature tends to reflect that semantic content - the signature is, in effect, "about" the same knowledge domain. In some cases, the cosine similarity reaches as high as 0.4 (e.g., social science knowledge benchmarks). On the other hand, some meta-ability benchmark signatures bear little relation to their intended functions, such as logical reasoning.

Why do some benchmark signatures "match" the stated function while others don't? We have three theories: (1) Benchmarks often bundle multiple subskills: beyond the target ability, they depend on instruction following, reading load, and format handling. As a result, signature tokens often reflect whichever auxiliary factor drives the most performance variance. Knowledge benchmarks are cleaner, while abstract meta-ability tasks (e.g., logical reasoning, detecting missing information) are more distorted by these side demands and by gaps between task design and implementation. (2) Signatures come from predictive token-level perplexity in natural corpora; when the intended skill is rare or procedural (like "logical reasoning", "detect missing information"), models default to proxy cues—genre, discourse markers, instruction tokens—rather than domain-specific features. This problem is smaller for well-defined knowledge areas. Also, signatures often include numerals, syntax tokens, or discourse markers that look semantically unrelated, whereas knowledge tasks appear more semantically aligned simply because their signatures form coherent domain narratives. (3) Strong predictive power doesn't imply shared semantics: models can rely on statistical co-occurrences in the natural corpora correlated with appearances of benchmark questions rather than true semantic relations. Semantic embeddings therefore cannot fully approximate models' internal task representations, consistent with findings of transferable but non-human-interpretable structures (Musker et al., 2025; Wu et al., 2024). Benchmark overlap here refers to how similarly models are confused by two sets of silent tokens - not to the semantic or textual overlap of the signature content. We have a list of representative benchmark signatures as shown in Appendix A.9.

## 6 FINAL REMARKS

LLM benchmark saturation has been widely discussed (Phan et al., 2025). Instead of introducing ever harder benchmarks, we propose benchmark signatures, a principled method to quantify overlap among LLM benchmarks. We ground benchmark relationships in cross-model perplexity patterns from in-the-wild corpora and compare them to surface semantics and correlated performance. We find signatures robust to benchmark-orthogonal factors (e.g., question format) while revealing both expected and unexpected cross-domain entanglements. Signatures are defined by the predictive power of tokens: tokens whose model perplexity patterns strongly predict benchmark outcomes, regardless of raw perplexity. Such tokens capture how structural properties of model training align with benchmark capability demands, rather than whether models have merely "seen" the required content. Our findings advance understanding of the LLM capacity space, benchmark validity, and model sensitivities. Future directions include extending signatures to finer-grained probes (e.g., layer-level activations and interpretability) and generalizing beyond QA or true–false tasks, such as open-ended generation (summarization, long-form reasoning, and dialogue) that requires stable, reproducible scoring functions. More work on causality would also be valuable. Broadly, our approach suggests a "benchmark algebra" for decomposing, recombining, and comparing benchmarks to expose gaps or redundancies, enabling the creation of entirely new benchmarks that target capabilities or failure modes identified through principled analysis. Together, these extensions position benchmark signatures as a reusable diagnostic toolkit for evaluating and improving benchmark ecosystems.

## THE USE OF LARGE LANGUAGE MODELS (LLMS)

We employed LLMs to assist with polishing the writing. All content generated or modified by LLMs was rigorously reviewed and approved by the authors.

## ETHICS STATEMENT

This work does not involve human subjects, sensitive data, or any other issues outlined in the ICLR Code of Ethics.

## REPRODUCIBILITY STATEMENT

To ensure the reproducibility of our experiments, we provide detailed descriptions of all methodologies in Sections 2 and 3. In addition, Appendix A.5 contains a walkthrough of each key checkpoint and experimental setup, including (but not limited to) important numerical values, evaluation metrics, and the software packages used for implementation.

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

# A  APPENDIX

## A.1  RELATED LITERATURE

**Benchmark Categorization and Overlap**: Benchmarks are central to model evaluation. Two simple metrics capture their utility: signal, a benchmark's ability to reliably distinguish better models from worse ones, and noise, a benchmark's sensitivity to randomness (Heineman et al., 2025). Recently, researchers have begun to ask how comparable benchmarks with similar intent actually are. This is commonly studied through Benchmark Agreement Testing (BAT), where new benchmarks are validated against established ones using agreement metrics (e.g., rank correlation) (Perlitz et al., 2024). Such analyses have led to concerns that the community may be producing too many benchmarks. For example, Liu et al. (Liu et al., 2021) examined agreement across multiple QA benchmarks and concluded that because agreement was high, additional QA benchmarks were unnecessary. Beyond statistical agreement, some recent works have attempted to qualitatively interpret and categorize benchmarks - for example, as testing logical reasoning or commonsense reasoning - though often without running agreement tests either within or across these categories (Ni et al., 2025). Recent human-curated benchmarks, such as "humanity's last exam", explicitly aim to mitigate saturation (Phan et al., 2025), while our work provides a mechanistic explanatory account of why existing benchmarks saturate and overlap in the first place. Another emerging line of inquiry asks what capabilities are still missing from current benchmark suites. Miller and Tang (Miller & Tang, 2025), for instance, examine how people commonly use LLMs for summarization, technical assistance, reviewing work, data structuring, generation, and information retrieval, and assess the extent to which existing benchmarks cover these capabilities. Their findings reveal significant gaps in coverage of benchmarks across categories.

**Signal Extraction from In-the-wild Data**: A growing body of work investigates how information extracted from in-the-wild corpora can inform data selection and model evaluation, even building benchmarks automatically. A central insight is that LLM losses on in-the-wild texts are often correlated with downstream benchmark performance, suggesting that simple loss-performance correlation coefficients can be effective signals for identifying high-quality training data from in-the-wild corpus (Thrush et al., 2025; Hoffmann et al., 2022). Validation loss is thus frequently used as a proxy for model generalization (Kaplan et al., 2020; Hoffmann et al., 2022; Wei et al., 2022), and with more recent evidence showing that such correlations persist across architectures and training settings (Poli et al., 2023). One line of research focuses on efficient, low-cost methods for understanding and filtering signals, for instance lightweight approaches using surface-level heuristics (n-gram overlap (Xie et al., 2023) or semantic-level similarities (Everaert & Potts, 2023)), enabling scalable filtering of massive corpora. Thrush et al., (Thrush et al., 2025) proposed an orthogonal approach for data selection centered around estimates of perplexity-benchmark correlations. We build on these ideas to construct benchmark signatures by mining predictive tokens of LLM performance from large-scale in-the-wild corpora, in order to address challenges in meta-evaluation - the evaluation of LLM evaluations, e.g., how overlapping they are.

## A.2  COMPARISONS BETWEEN TOKEN-, CHUNK-, AND DOCUMENT-LEVEL PERPLEXITY

From fine to coarse granularity, we consider token-, chunk-, and document-level perplexities. At the **document level**, we evaluate the model on an entire document and take the mean across all text chunks that fit within the model's context window (part of the document). At the **chunk level**, we split documents into fixed-length windows (30 pieces, using spaces as separators) and compute perplexity as the average over all tokens within each window. At the **token level**—the finest granularity with the least inductive bias—we use token-wise perplexities from documents to capture the model's intrinsic uncertainty. Concretely, we form a window by taking the target token with its up-to-30 preceding pieces (using spaces as separators) as context, then record only the last token's perplexity as the feature. This ensures the token is conditioned on its preceding context rather than treated in isolation. As shown in Table 1, the *token* level exhibits the greatest standard deviation and interquartile range, as well as more pronounced extreme gaps in majority cases compared to the *chunk* and *doc* levels. This wider dispersion indicates that extreme values are more visible and significant at the token level, making it a natural choice for feature selection. Token-level signatures balance the strongest predictive power (both positive and negative relations) of highly informative tokens, and they exhibit high variance of predictive power, as captured by the deviations. By focusing on the

token level, we are able to highlight more prominent signals, whereas aggregation at the chunk or document level tends to smooth out these extremes.

| Benchmark | Level | Std | IQR | Max–Q99 | Q01–Min | $R^2_{adj}$ |
|---|---|---|---|---|---|---|
| Gsm8k | Chunk | 30.30 | 44.63 | **43.37** | **36.63** | 0.903 |
| Gsm8k | Doc | 19.39 | 26.50 | 23.25 | 18.75 | 0.826 |
| Gsm8k | Token | **36.53** | **54.11** | 40.63 | 27.37 | **0.927** |
| Mbpp | Chunk | 29.83 | 43.05 | **87.62** | 27.05 | 0.849 |
| Mbpp | Doc | 14.49 | 18.78 | 32.44 | 8.67 | 0.667 |
| Mbpp | Token | **36.76** | **50.60** | 39.60 | **39.20** | **0.885** |
| Mmlu | Chunk | 29.31 | 41.18 | **64.00** | 30.36 | **0.551** |
| Mmlu | Doc | 18.11 | 27.68 | 35.16 | 9.68 | 0.185 |
| Mmlu | Token | **38.05** | **51.90** | 52.00 | **42.29** | 0.454 |
| Truthfulqa | Chunk | 30.58 | 41.80 | 48.00 | 29.20 | 0.414 |
| Truthfulqa | Doc | 17.67 | 22.75 | 25.00 | 13.00 | 0.208 |
| Truthfulqa | Token | **36.57** | **53.00** | **49.20** | **35.40** | **0.505** |

Table 1: **Summary of `Thrush` coefficient distributions across four benchmarks.** The columns report standard deviation (Std), interquartile range (IQR), and tail gaps of Max-Q99 and Q01-Min, which are defined as the distance from the maximum to the 99th percentile (Max–99th) and from the 1st percentile to the minimum (1st–Min). Adjusted Coefficient of Determination ($R^2_{adj}$) is extracted from the actual fit of the linear model across different granularities. Across 20 targets (5 measures × 4 benchmarks), token-level values achieved 15 wins. For the five losses, chunk-level statistics perform slightly better. This is because chunk-level distributions contain more outliers, meaning that the 1st and 99th percentile values can be extremely low or high (where they win), while the standard deviation is not as pronounced as that in the token-level case. We thus mainly rely on Std and IQR for the final selection. Note that our framework is conceptually extendable to chunk-level and document-level measures. Token-level measures also introduce the least inductive bias (minimal structural assumptions and segmentation artifacts) while offering the highest granularity and more faithful representation of model uncertainty.

## A.3 CONDITIONS FOR SURE INDEPENDENCE SCREENING (SIS)

Sure Independence Screening (SIS) is a powerful statistical tool for feature selection in ultra-high dimensional settings, offering a "sure screening property" that guarantees the retention of truly informative features with high probability under specific conditions (Fan & Lv, 2008). In this section, we elaborate on how the key theoretical assumptions underlying SIS are plausibly met within our problem context of mining benchmark signatures from token-level perplexities.

1. **Ultra-High Dimensionality:** Our problem inherently operates in an ultra-high dimensional regime, where the number of "in-the-wild" tokens ($d$, scaling to billions) vastly exceeds the number of language models ($m$, typically in the tens). Specifically, we have $\log(d) > m$, which far exceeds the standard $d > m$ high-dimensional definition. This extreme disparity makes full multivariate regression computationally intractable, underscoring the necessity of an efficient screening step like the one we employ.

2. **Sparsity:** The "Sparsity" assumption (our **[A1]**) posits that only a small fraction of the $d$ tokens are truly informative for predicting LLM benchmark performance. Our empirical observations of the correlation coefficient distributions (e.g., Figure 3) directly support this. The distributions show a strong concentration around zero, indicating that most tokens have little to no marginal predictive power. The presence of thin but distinct tails also suggests that a small subset of tokens exhibits strong correlations, aligning with the idea that specific linguistic phenomena (represented by these tokens) drive performance on a given benchmark.

3. **Minimum Signal Strength:** SIS requires that the true predictive signals (i.e., the tokens with non-zero effects on benchmark performance) are not arbitrarily weak. In our context, this translates to these important tokens having sufficiently strong marginal correlations to stand out from the noise. Our use of token-level perplexities, which directly reflect an LLM's familiarity of specific linguistic patterns, suggests that truly important tokens would indeed manifest as strong signals. The robust, rank-based correlation coefficients we employ (`Thrush` and

`Pre-select`) are also well-suited to detect such signals, as they are less sensitive to outliers and distributional peculiarities that might obscure signals when using less robust measures.

4. **Limited Pathological Multicollinearity:** A critical condition for basic SIS is that the multi-collinearity between important features and unimportant ones should not be so severe that it masks the marginal signal of truly predictive tokens (e.g., the suppressor variable scenario). While token perplexities can exhibit correlations (e.g., highly similar tokens or tokens from common linguistic constructs), it is less probable that a truly causal token's signal would be perfectly canceled out by others at a marginal level. Benchmarks typically probe specific abilities, which are likely associated with a distinct, though perhaps overlapping, set of "signature" tokens. The vast and diverse nature of "in-the-wild" tokens also means that while many tokens might be highly correlated, there are many more effectively independent ones. More importantly, the core objective of our work is to identify benchmark signatures as a specific and parsimonious set of tokens. If a token's marginal signal is entirely masked, it might suggest its contribution is highly redundant with other tokens that do have a strong marginal signal, or that its unique contribution is extremely weak – in which case, its exclusion from the initial screening might not significantly harm the final signature's predictive power.

## A.4 TECHNICAL DETAILS

### A.4.1 SEMANTIC-LEVEL BOOTSTRAPPED SIMILARITY CALCULATION

---

**Algorithm 2** Get Pairwise Similarity Matrix

---

**Input:** A list of benchmarks $\mathcal{B} = \{B_1, \ldots, B_n\}$; Embedding model $E$ (e.g. a sentence transformer); Number of bootstrap replicates $k$.
**Output:** An $n \times n$ similarity matrix $S$.

1: $n \leftarrow |\mathcal{B}|$
2: $S \leftarrow$ an $n \times n$ matrix initialized to zeros
3: **for** $i = 1$ **to** $n$ **do**
4:     **for** $j = i$ **to** $n$ **do**
5:         **if** $i = j$ **then**
6:             $S_{i,j} \leftarrow 1.0$
7:         **else**
8:             $s \leftarrow$ getSimScore$(B_i, B_j, E, k)$   ▷ Call Algorithm 3
9:             $S_{i,j} \leftarrow s$
10:            $S_{j,i} \leftarrow s$  ▷ Matrix is symmetric
11:         **end if**
12:     **end for**
13: **end for**
14: **return** $S$

---

---

**Algorithm 3** Bootstrapped Similarity Score Calculation (getSimScore)

---

**Input:** Benchmarks $A$ and $B$; Embedding model $E$; Bootstrap replicates $k$.
**Output:** A single similarity score $\text{sim}_{A,B}$.
  1: ▷ Determine which benchmark has a smaller size
  2: **if** $|A| < |B|$ **then**
  3:     $S \leftarrow A; L \leftarrow B$ ▷ $S$ is the smaller, $L$ is the larger
  4: **else**
  5:     $S \leftarrow B; L \leftarrow A$
  6: **end if**
  7: $n_s \leftarrow |S|$ ▷ Get the size of the smaller benchmark
  8: $\ell \leftarrow \text{getMaxLength}(E)$ ▷ Obtain the maximum processing length
  9: ▷ Process the smaller benchmark to get its single embedding
10: $\text{text}_S \leftarrow$ concatenate all questions in $S$
11: $\text{text}'_S \leftarrow \text{truncate}(\text{text}_S, \ell)$
12: $\text{emb}_S \leftarrow \text{encode}(E, \text{text}_S^{\text{trunc}})$
13: ▷ Generate bootstrap samples from the larger benchmark
14: $\mathcal{T}_L \leftarrow$ an empty list
15: **for** $b = 1$ **to** $k$ **do**
16:     $L' \leftarrow \text{sample}(L, n_s, \text{replace} = \text{False})$
17:     $\text{text}_L \leftarrow$ concatenate all questions in $L'$
18:     $\text{text}'_L \leftarrow \text{truncate}(\text{text}_L, \ell)$
19:     Append $\text{text}'_L$ to $\mathcal{T}_L$
20: **end for**
21: ▷ Batch-encode all samples and compute average similarity
22: $\text{embs}_L \leftarrow \text{batchEncode}(E, \mathcal{T}_L)$
23: $\text{similarities} \leftarrow \text{cosineSimilarity}(\text{emb}_S, \text{embs}_L)$ ▷ One-vs-many comparison
24: $\text{sim}_{A,B} \leftarrow \text{average}(\text{similarities})$
25: **return** $\text{sim}_{A,B}$

---

### A.4.2   AIC STEPWISE FORWARD SELECTION ALGORITHM

---

**Algorithm 4** Selecting Salient Tokens for Benchmark $j$

---

**Input:** Perplexity feature matrix $\mathbf{P}$; performance vector $\mathbf{y}_{:,j}$; tail fraction $\alpha = 0.01$; tolerance $\delta \geq 0$
**Output:** Salient Token set $\mathcal{T}'_j$
  1: **Preselection via Thrush Correlation**
  2: **for** $\ell = 1$ to $d$ **do**
  3:     $\rho_\ell \leftarrow \text{ThrushCorr}(\mathbf{P}_{:,\ell}, \mathbf{y}_{:,j})$
  4: **end for**
  5: $T^+ \leftarrow$ indices of the top $\alpha d$ values of $\rho_\ell$ ▷ most positively correlated
  6: $T^- \leftarrow$ indices of the bottom $\alpha d$ values of $\rho_\ell$ ▷ most negatively correlated
  7: $T \leftarrow \text{Shuffle}(T^+ \cup T^-)$ ▷ candidate feature set
  8: **Forward Selection with AIC (on $T$)**
  9: $S \leftarrow \emptyset; \quad A^\star \leftarrow +\infty$
10: **while** $T \setminus S \neq \emptyset$ **do**
11:     **for** each $\ell \in (T \setminus S)$ **do**
12:         $A(\ell) \leftarrow \text{AIC}\big(\text{Fit}(\mathbf{y}_{:,j} \sim \mathbf{P}_{:,\, S\cup\{\ell\}})\big)$
13:     **end for**
14:     $\ell^\star \leftarrow \arg\min_{\ell\in(T\setminus S)} A(\ell); \quad A_{\text{new}} \leftarrow A(\ell^\star)$
15:     **if** $A_{\text{new}} < A^\star - \delta$ **then**
16:         $S \leftarrow S \cup \{\ell^\star\}; \quad A^\star \leftarrow A_{\text{new}}$
17:     **else**
18:         **break** ▷ no further AIC improvement
19:     **end if**
20: **end while**
21: $\mathcal{T}'_j \leftarrow S$
22: **return** $\mathcal{T}'_j$

---

## A.5 EXPERIMENT SETUP

**Overview**: Our chosen benchmarks span diverse domains such as knowledge (business, humanities, social sciences, science and engineering, medicine), mathematics, coding, reasoning, language, culture and world knowledge, logic, and instruction following. We choose 32 widely-used language models (see the list below). We extract benchmark signatures from the open dataset *RedPajama* (Weber et al., 2024), which contains large-scale textual data across multiple domains, including CommonCrawl, C4, GitHub, arXiv, Books, Wikipedia, and StackExchange, used for pretraining LLMs, making it a strong source of in-the-wild data for mining benchmark signatures. We take the standard approach, using vLLM (Kwon et al., 2023) for facilitating perplexity extraction and `llm-evaluation-harness` (Gao et al., 2024) for evaluation across benchmarks and models such that all evaluations are under the same condition.

### A.5.1 EXPERIMENT WALKTHROUGH

As discussed, we measure perplexity at three granularities - token, chunk, and document levels (from fine to coarse). The segmentation procedure for each is detailed in §A.2. We ultimately focus on the *token level* because it provides the clearest view of prominent signals for the pre-filtering stage.

**Preprocessing RedPajama**    We use the 1B-token RedPajama variant to balance scale and computational cost. For token-level segmentation, we split the corpus on whitespace into pieces. For each piece, we prefix up to the preceding 30 pieces as left context and record the *last token's* perplexity conditioned on that context. This yields an initial pool on the scale of billions of token-level contexts ($d \approx 8.45 \times 10^9$). To reduce noise or in-the-wild text, we uniformly downsample by a factor of $1/50$, yielding approximately $1.69 \times 10^7$ instances.

**Feature Matrix Construction**    Using the vLLM setup described in §A.5.4, we evaluate *32 models* on the token contexts and extract token-level perplexities, forming the covariate (feature) matrix

$$\mathbf{P} \in \mathbb{R}^{32 \times 1.69 \times 10^7},$$

with rows indexed by models and columns by token instances.

**Performance Matrix Construction**    In parallel, we compute model performance on a series of benchmarks and subfields using the `lm-evaluation-harness` (details in §A.5.5). Let

$$\mathbf{Y} \in \mathbb{R}^{32 \times 89}$$

denote the performance matrix (models × benchmarks/subfields). For each benchmark $B_j$, the vector $\mathbf{y}_{:,j}$ is the *performance vector* for $B_j$ across all 32 models.

**Filtering with Thrush**    For each benchmark $B_j$, we compute the Thrush rank correlation between the entire feature matrix $\mathbf{P}$ and the performance vector $\mathbf{y}_{:,j}$. This produces a distribution of Thrush scores over token features. We retain the top 1% and bottom 1% features (by score) and concatenate these extremes into a benchmark-specific subset of columns from $\mathbf{P}$ for downstream modeling.

**AIC Step-Forward Feature Selection**    Finally, for each benchmark $B_j$, we fit a multivariate linear model on the preselected features using step-forward selection with the Akaike Information Criterion (AIC) as the objective. Starting from an empty model, we iteratively add the feature that most improves AIC and stop when no further improvement is possible (tolerance = 0). The resulting selected set constitutes the most predictive in-the-wild token features for $B_j$. Across benchmarks, the selected set size varies but typically has $\sim 30$ features.

**Signature and Comparison**    Consider two benchmark signature vectors, $S_1$ and $S_2$, each consisting of several context pieces (30 pieces separated by spaces) plus the salient token. For each benchmark we acquire around 30 such non-overlapping salient tokens. We evaluate these signatures with 32 models, which read their respective pre-contexts and compute last-token perplexities. If the models exhibit similar levels of perplexity for both signatures, this strongly suggests that the two benchmarks align. We normalize each model's perplexity values into their $z$-score within the model. For each model, we then compute the mean of the $z$-scored perplexities for the two benchmark signatures. Finally, we calculate the Spearman correlation ($\rho_s$) between these two mean vectors to represent signature-level overlap.

## A.5.2 DATASETS

| Benchmark family | Benchmarks |
| --- | --- |
| mmlu | business_ethics; marketing; management; professional_accounting; high_school_european_history; jurisprudence; humanities; prehistory; professional_law; world_religions; high_school_us_history; formal_logic; high_school_world_history; international_law; logical_fallacies; moral_disputes; moral_scenarios; philosophy; anatomy; clinical_knowledge; college_medicine; human_aging; medical_genetics; nutrition; professional_medicine; virology; miscellaneous; other; abstract_algebra; astronomy; college_biology; college_chemistry; college_physics; conceptual_physics; elementary_mathematics; high_school_biology; high_school_computer_science; college_computer_science; college_mathematics; computer_security; electrical_engineering; high_school_chemistry; high_school_mathematics; high_school_physics; high_school_statistics; machine_learning; stem; high_school_geography; high_school_government_and_politics; high_school_macroeconomics; high_school_microeconomics; high_school_psychology; public_relations; us_foreign_policy; econometrics; human_sexuality; professional_psychology; security_studies; social_sciences; sociology |
| bbh | global_facts; hyperbaton; snarks; web_of_lies; word_sorting; disambiguation_qa; salient_translation_error_detection; boolean_expressions; dyck_languages; geometric_shapes; multistep_arithmetic_two; navigate; object_counting; reasoning_about_colored_objects; formal_fallacies; logical_deduction_five_objects; logical_deduction_seven_objects; logical_deduction_three_objects; penguins_in_a_table; temporal_sequences; tracking_shuffled_objects_five_objects; tracking_shuffled_objects_seven_objects; tracking_shuffled_objects_three_objects; causal_judgement; movie_recommendation; ruin_names; date_understanding; sports_understanding |

Table 2: Benchmarks in MMLU and BBH.

| Format | Benchmarks |
| --- | --- |
| multi-choice questions | business_ethics; marketing; management; professional_accounting; high_school_european_history; jurisprudence; humanities; prehistory; professional_law; world_religions; high_school_us_history; formal_logic; high_school_world_history; international_law; logical_fallacies; moral_disputes; moral_scenarios; philosophy; anatomy; clinical_knowledge; college_medicine; human_aging; medical_genetics; nutrition; professional_medicine; virology; miscellaneous; other; abstract_algebra; astronomy; college_biology; college_chemistry; college_physics; conceptual_physics; elementary_mathematics; high_school_biology; high_school_computer_science; college_computer_science; college_mathematics; computer_security; electrical_engineering; high_school_chemistry; high_school_mathematics; high_school_physics; high_school_statistics; machine_learning; stem; high_school_geography; high_school_government_and_politics; high_school_macroeconomics; high_school_microeconomics; high_school_psychology; public_relations; us_foreign_policy; econometrics; human_sexuality; professional_psychology; security_studies; social_sciences; sociology |
| true or false | boolean_expressions; causal_judgement; formal_fallacies; navigate; sports_understanding; web_of_lies |

Table 3: Classification of benchmarks by question format.

A.5.3 MODELS USED

The models we used in this study are summarized in Table 4. Each entry cites the official paper if available, otherwise the model card.

| Model (HF Repository) | Citation |
| --- | --- |
| meta-llama/Llama-3.1-8B-Instruct | Grattafiori et al. (2024) |
| meta-llama/Llama-3.2-1B-Instruct | Grattafiori et al. (2024) |
| meta-llama/Llama-3.2-3B-Instruct | Grattafiori et al. (2024) |
| google/gemma-3-4b-it | Team (2025a) |
| google/gemma-3-12b-it | Team (2025a) |
| google/gemma-3-27b-it | Team (2025a) |
| mistralai/Mistral-7B-Instruct-v0.3 | Jiang et al. (2023) |
| deepseek-ai/deepseek-llm-7b-chat | DeepSeek-AI et al. (2024) |
| Qwen/Qwen3-0.6B | Team (2025b) |
| Qwen/Qwen3-1.7B | Team (2025b) |
| Qwen/Qwen3-4B | Team (2025b) |
| Qwen/Qwen3-8B | Team (2025b) |
| tiiuae/falcon-rw-1b | Penedo et al. (2023) |
| EleutherAI/pythia-1b | Biderman et al. (2023) |
| EleutherAI/pythia-6.9b-v0 | Biderman et al. (2023) |
| EleutherAI/pythia-12b-deduped | Biderman et al. (2023) |
| mosaicml/mpt-7b-instruct | Team (2023) |
| microsoft/Phi-3-mini-4k-instruct | Abdin et al. (2024a) |
| microsoft/Phi-4-mini-instruct | Microsoft et al. (2025) |
| microsoft/Phi-4 | Abdin et al. (2024b) |
| 01-ai/Yi-1.5-9B-Chat | AI et al. (2025) |
| 01-ai/Yi-1.5-6B-Chat | AI et al. (2025) |
| mistralai/Ministral-8B-Instruct-2410 | Jiang et al. (2024) |
| openai-community/gpt2-medium | Radford et al. (2019) |
| openai-community/gpt2-large | Radford et al. (2019) |
| openai-community/gpt2-xl | Radford et al. (2019) |
| zai-org/chatglm3-6b | GLM et al. (2024) |
| zai-org/glm-4-9b-chat-hf | GLM et al. (2024) |
| zai-org/codegeex4-all-9b | Zheng et al. (2023) |
| allenai/OLMo-2-1124-13B-Instruct | OLMo et al. (2024) |
| allenai/OLMo-2-1124-7B-Instruct | OLMo et al. (2024) |
| allenai/OLMo-2-0425-1B-Instruct | OLMo et al. (2024) |

Table 4: Models used in this study. Multiple variants may share the same citation.

A.5.4 INFERENCE WITH VLLM

We use `vLLM` (Kwon et al., 2023) for facilitating perplexity extraction. Specifically, we run all inference in offline mode with tensor parallelism across 2 GPUs to maximize throughput and data parallelism across 1 GPU. The model weights are cached locally to reduce repeated I/O overhead, and inference is performed in batches of prompts to further amortize the computation cost. For each sequence, we extract per-token log probabilities, from which we compute negative log-likelihood and perplexity metrics. All outputs are aggregated into parquet for downstream analysis. The GPUs we used in this computation are $2 \times$ Nvidia A100 (80GB).

A.5.5 BENCHMARK EVALUATION AND LABELING

For benchmark evaluation we use `llm-evalution-harness` (Gao et al., 2024). Each benchmark involves different task formats, and we adopt the standard metrics defined in the harness to ensure comparability. For MMLU, which consists of multi-choice questions across 57 academic subjects, we report *accuracy*, i.e., the proportion of questions with correctly selected options. For MBPP, a code generation benchmark, we evaluate using pass@1, the fraction of problems solved

correctly on the first attempt, based on unit test execution. For BBH, which is a collection of heterogeneous tasks (multiple-choice, binary classification, and completion), we follow the harness in applying the canonical metric for each benchmark. We use accuracy for multiple-choice and true/false items, and exact match for sentence completions. For IFEval, which tests instruction-following, we adopt the harness's compliance accuracy, quantifying the percentage of model responses that satisfy the explicit constraints in the prompt. These heterogeneous metrics reflect the intended difficulty and modality of each benchmark, and together provide a broad view of model capability. For AbsenceBench (Fu et al., 2025), we use the average scores across three dimensions: numerical, poetry, and GitHub.

For each benchmark we follow these rules to label its function:

- If it's about math problems, then we label it as "mathematics", including MMLU abstract algebra, elementary mathematics, college mathematics, high school mathematics, and high school statistics (Hendrycks et al., 2021).
- Coding — drawing from the MBPP benchmark (Austin et al., 2021).
- Instruction Following — drawing from the IFEval benchmark (Zhou et al., 2023).
- Scientific Knowledge — MMLU domains such as business, humanities, natural science and engineering, social sciences, and medicine.
- Language — (BBH) semantic understanding, name disambiguation, entity resolution, grammar rules, and sarcasm detection.
- World Knowledge — (BBH) cultural and general world knowledge, including common practices and presuppositions (mostly) in Western society. Examples of world knowledge tasks include the following: Sports Understanding, Movie Recommendation, and Date Understanding.
- Logic (Formal Logic) — abstract study of propositions/statements and deductive arguments (e.g., MMLU's formal logic and logical fallacies).
- Reasoning — (BBH) tasks spanning arithmetic (e.g., multi-step arithmetic), logical structures (e.g., Boolean expressions, deduction), geometric (e.g., geometric shapes), hierarchical (e.g., Dyck languages), spatial (e.g., navigation), and temporal (e.g., temporal sequences).
- AbsenceBench (Fu et al., 2025) - the ability to tell what's missing.
- Note that we mostly refer directly to the official labels (e.g., what falls under "reasoning", "world knowledge", etc.) given in the official article of Suzgun et al. (2022) (section 5) without making changes.

## A.6 STATISTICAL ANALYSIS OF BENCHMARK RELATIONS

As shown in Table 5, we used a bootstrapping approach to evaluate whether the signature correlations within a benchmark category differ statistically from those across categories. For each pair of benchmarks, we computed the overlap between their signatures, which reflects how similarly the chosen set of representative LLMs are "confused" by the two benchmarks (details see Section 3.3 in the main text). Because the numbers of within-category and cross-category pairs differ, we performed 10,000 bootstrap samples to estimate the distributions and corresponding p-values. The resulting difference, expressed as a positive or negative percentile value, indicates how much larger or smaller the within-category mean correlations are compared to the mean cross-category correlation, and whether this deviation is statistically significant. Results are discussed in the main text Section 4-1.

## A.7 ROBUSTNESS ANALYSIS OF DESIGN, METHODS, PARAMETERS, AND CORPORA

### A.7.1 ROBUSTNESS ANALYSIS OF DESIGN

**Leave-One-Out Cross-Validation (LOOCV)** To assess generalization of the proposed framework, we performed LOOCV over 32 models on the 27 BBH sub-tasks, comparing our predictor to a baseline that uses the mean to predict the held-out model's performance. Our model achieved an

| Benchmark Category | Difference | P-value |
|---|---|---|
| Humanities | -46% | 0.00 |
| Reasoning | +21% | 0.00 |
| Language | -10% | 0.15 |
| Medicine | +6% | 0.54 |
| Science and Engineering | +42% | 0.00 |
| Social Science | +41% | 0.00 |
| World Knowledge | -17% | 0.01 |
| Business | +2% | 0.70 |

Table 5: Within- and cross-category benchmark differences.

MAE of $0.076 \pm 0.038$, substantially better than the baseline's $0.181 \pm 0.144$. This confirms that the extracted features capture genuine, reusable structure rather than overfitting. We further tested robustness with a strict out-of-sample evaluation using two fully held-out models – `Qwen2.5-7B` and `Falcon-rw-7b`. Using the 27 training models to derive features and forecast performance across 27 benchmarks, we obtained MAEs of $0.0681 \pm 0.034$ and $0.0722 \pm 0.026$, respectively. These results show that the model generalizes well even to unseen observations.

**Confounds of the "Base" Ability** To empirically adjudicate whether our signatures capture distinct, subject-specific factors versus a generic "dataset style" (e.g., a common "factor" shared across all BBH/Wikipedia tasks), we conducted a Cross-Task Transfer Experiment. If the alternative hypothesis is correct — that signatures primarily capture a generic capability or dataset artifact — a signature derived from Benchmark $A$ ($S_A$) should successfully predict the performance on a different Benchmark $B$ ($Y_B$), provided they share that generic source.

*Experiment Setup:*

We performed the following for all 27 BBH subtasks: For a given benchmark $A$, we fixed its covariate feature matrix from its signatures $X_A$. We then trained 26 separate regression models, each regressing the performance vector of a different benchmark $Y_B$ (where $B \neq A$) against $X_A$ and collecting the absolute error of the prediction on $B$ from $X_A$ against the true value of $Y_B$. We iterated this process for all $A$ (all subtasks in BBH), resulting in a comprehensive evaluation of how well one task's signature predicts another task's performance. We also refit $X_B$ vs. $Y_B$ again and extract the absolute error each time to obtain the task-specific baseline.

*Results:*

- Baseline (predict based on mean; LOOCV from the general response): $0.181 \pm 0.144$
- Cross-Task Prediction: $0.223 \pm 0.048$
- Task-specific baseline: $0.021 \pm 0.012$

The observed MAE is significantly larger when attempting to use one task's signature to predict another compared to using the task's own signature (0.223 vs. 0.021). Crucially, the cross-task error is even higher than the naive LOOCV baseline (0.181), where we simply used the mean of the performance vector. If the signatures merely encoded a generic "style" or shared data contamination, the covariates of Task $A$ ($X_A$) would serve as a sufficient proxy for the capabilities required by Task $B$ ($Y_B$). Under this hypothesis, training on $X_A$ to predict $Y_B$ should yield a good fit. The observed degradation in performance contradicts this, indicating that $X_A$ encodes specific, non-transferable structural information unique to Task $A$.

A.7.2 ROBUSTNESS ANALYSIS OF METHODS

**Ablation Study of Regularization Methods** We ran an ablation study replacing AIC-based forward selection (abbreviated as "AIC" below) with Lasso ($\alpha = 1$) and Elastic Net ($\alpha = 1$, $l_1$ratio $= 0.5$), keeping all other regression settings fixed. For each MMLU benchmark, we derived a signature vector under each method and computed the corresponding inter-benchmark correlation matrix. We then flattened each matrix's upper triangle and measured Spearman correlations ($\rho$)

among regression methods to assess whether the structural relationships among benchmarks persist under different regularizers. Results are:

- AIC vs. Lasso $\rho = 0.763$
- AIC vs. Elastic Net $\rho = 0.765$
- Lasso vs. Elastic Net $\rho = 0.786$

As a baseline, using 50 randomly sampled features (after the correlation filtering) produced $\rho = 0.334$ against AIC. These give us two interesting insights:

1. As expected, the initial filtering helps retain (marginally) informative tokens, so signatures constructed from random sampling have a non-trivial but weak correlation with regression-based counterparts.

2. The moderate-to-strong correlations among regression methods show that while the chosen features may vary, the between-benchmark structural relationships revealed by the signatures remain largely stable across regularization strategies.

**Spearman Correlation or Mutual Information for Screening:**    We now test alternative selection methods in the preselection phase. To validate our choice, we compared Thrush against Spearman correlation and Mutual Information (MI). We normalized all scores to a $[0, 1]$ scale and analyzed the standard deviation (Std) of their distributions across benchmarks. Our analysis reveals that Mutual Information consistently exhibits a lower Std, indicating significantly lower discriminative power compared to the ranking metrics.

| Benchmark | Metric | Std (0–1 Scale) |
|---|---|---|
| MBPP | thrush | 0.1549 |
| MBPP | spearman | 0.1562 |
| MBPP | mutual_info | 0.1057 |
| IFEVAL | thrush | 0.1557 |
| IFEVAL | spearman | 0.1557 |
| IFEVAL | mutual_info | 0.0926 |
| BBH | thrush | 0.1808 |
| BBH | spearman | 0.1808 |
| BBH | mutual_info | 0.0981 |

Furthermore, when comparing Thrush and Spearman, we found they produce nearly identical selections, with a $99.5\%$ overlap in selected features. Given this equivalence and its high discriminative capability, we maintained the use of Thrush.

### A.7.3    ROBUSTNESS ANALYSIS OF DATA SELECTION

While it is true that our signatures are extracted from RedPajama, this does not undermine their relevance.

First, RedPajama is broadly representative of "in-the-wild" web data including Wikipedia, GitHub, C4, arxiv, etc, which forms the dominant component of modern LLM pretraining. As noted in (Wolfram & Schein, 2025), LLMs are increasingly converging because major model families are trained on similar mixtures of large-scale web corpora, code, and curated text. In other words, although individual datasets differ at the margins, they share substantial structural and statistical overlap. Crucially, our goal is not to reconstruct or identify the exact training data of any model. Rather, we aim to capture generalizable distributional signatures that emerge across large in-the-wild corpora. Because RedPajama reflects the broad characteristics of public web text – and because model training corpora largely draw from the same underlying data universe – RedPajama provides a sufficiently representative substrate for extracting robust signatures. Thus, the method does not depend on exact training data matching. Instead, it leverages the empirical regularities of large-scale in-the-wild text, which are shared across most contemporary LLMs.

Second, to validate the robustness of our signatures against training data variations, we conducted a control experiment using Dolma (Soldaini et al., 2024), another massive training dataset derived

from diverse sources (including C4, arXiv, and others). We replicated our exact pipeline on Dolma: preprocessing, downsampling, and extracting perplexities to generate signatures for all BBH sub-tasks and computing the correlation matrix that captures the inter-correlation between subtasks. By flattening the upper triangles of the matrices and calculating the Spearman correlation between the matrices derived from RedPajama and Dolma, we obtained a high agreement of 0.895. This strong correlation demonstrates that the task signatures are robust to the specific choice of corpus, provided that the data is a sufficiently large and representative sample of in-the-wild data.

### A.7.4 ROBUSTNESS ANALYSIS OF PARAMETER CHOICES

**Motivation for the 1% pre-filtering threshold**

The pre-filtering step is guided by both statistical and computational considerations.

**Statistically.** The distribution of robust feature–outcome correlations in our dataset is approximately bell-shaped (see Fig 3). The 1% threshold (capturing the top/bottom tails) is a conservative heuristic designed to encompass the heavy-tailed components (roughly $> 2.3$ standard deviations under a normal approximation) where the signal appears concentrated.

**Computationally.** The subsequent regression step scales as $\mathcal{O}(md'^2)$, where $d'$ is the number of features after pre-filtering. Applying a 1% cut reduces dimensionality by roughly two orders of magnitude, ensuring that the second stage remains tractable.

The overall idea is that thresholds substantially above 1% make the pipeline computationally difficult, while thresholds that are too small risk filtering away important features.

To demonstrate that our chosen preselect ratio is a **robust parameter** rather than an arbitrary choice, we performed a fine-grained sensitivity analysis on the BBH dataset. We examined the structural evolution of the model's understanding across a geometric grid of ratios

$$r \in \{10^{-6}, \ldots, 1.0\}.$$

**Experiment Setup**

For each ratio in the grid, we generated a $27 \times 27$ inter-benchmark correlation matrix (heat matrix) representing the pairwise relationships between tasks. To quantify structural stability, we compared the heat matrix at ratio $r_i$ to that at the next increment $r_{i+1}$. We flattened the upper triangle of each matrix and computed the Pearson correlation ($\rho$) between consecutive steps.

**Results**

The table below tracks the stability of the heat matrices. Each value represents the correlation between matrices constructed under $r_i$ and $r_{i+1}$.

| Grid Transition ($r_i \to r_{i+1}$) | Heat Matrix Correlation ($\rho$) |
|---|---|
| $10^{-6} \to 10^{-5}$ | 0.2429 |
| $10^{-5} \to 10^{-4}$ | 0.3923 |
| $10^{-4} \to 10^{-3}$ | 0.6310 |
| $10^{-3} \to 0.01$ | 0.9143 |
| $0.01 \to 0.1$ | 0.9837 |
| $0.1 \to 0.2$ | 0.9990 |
| $0.2 \to 0.3$ | 1.0000 |
| $0.3 \to 0.4$ | 1.0000 |
| $0.4 \to 0.5$ | 1.0000 |
| $0.5 \to 1.0$ | 1.0000 |

**Conclusion**

Visual inspection of the heat matrices combined with the quantitative correlation analysis reveals a clear phase transition. At low ratios ($r < 10^{-3}$), benchmark relationships are volatile. The structure stabilizes significantly by $r = 0.01$ ($\rho > 0.91$) and effectively converges by $r = 0.1$ ($\rho > 0.98$). Beyond $r = 0.2$, the heat matrices become identical ($\rho \approx 1.0$), confirming that selecting a ratio between 0.01 and 0.1 efficiently captures the stable, intrinsic structure of the BBH tasks without

requiring full feature saturation. The reduction in computational complexity substantially outweighs any residual sensitivity to the parameter choice.

## A.8    ANALYSIS OF COMPUTATIONAL COST

Our analysis in this section is structured in two parts. We first explain the computational cost, and then use experiments to show why our method is **less vulnerable to the scalability challenge**.

**Computational Cost and Scalability.**    Admittedly, we face computational constraints similar to prior work such as LESS (Xia et al., 2024). Because Stepwise AIC scales linearly with sample size ($N$), processing larger corpora becomes difficult. At 1B tokens, perplexity extraction takes approximately 2 hours per model, followed by 30–40 minutes of pre-filtering and regression per benchmark. Scaling to 10B tokens would increase these times to roughly 20 hours for extraction and 5 hours per benchmark for regression. This renders 100B+ scales computationally infeasible under our setting.

However, we found that 1B tokens are sufficient to capture robust structural relationships among the benchmarks. Our experiments below, which vary the token count from 100K to 10B ($10\times$ the original scale), clearly demonstrate that the stability of our method exhibits strong diminishing returns beyond the 1B-token threshold.

**Stabilization Analysis.**    Using a fixed 1% pre-selection ratio, we sampled corpora at logarithmic intervals (100K–10B) and computed task-to-task correlation matrices for the 27 BBH sub-tasks at each scale. Stability was measured via Spearman correlation between consecutive scales:

- 100K vs. 1M: $\rho = 0.278$
- 1M vs. 10M: $\rho = 0.187$
- 10M vs. 100M: $\rho = 0.475$
- 100M vs. 1B: $\rho = 0.998$
- 1B vs. 10B: $\rho = 1.000$

**Conclusion on Scalability.**    The results above demonstrate a "phase transition": signatures are unstable below 100M tokens but converge ($\rho > 0.99$) in the 100M–1B range. Since 1B tokens act as a sufficient statistical proxy, we can use a manageable subset rather than scaling linearly to trillions of tokens. The empirical study further shows that appropriate down-sampling yields nearly identical results while saving approximately an order of magnitude in computational resources.

The results also indicate that 100M tokens are empirically sufficient to capture signatures without incurring large-scale regression costs. Although our study employed a 1B-token sample, appropriate down-sampling remains feasible for researchers operating under computational constraints who wish to reproduce our findings.

## A.9    EXAMPLES OF BENCHMARK SIGNATURES

We show benchmark example signatures and coefficients of their predictive power for benchmark performance. The last part of each text (within '[]') is the silent token with the coefficient in parentheses. They are all high-predictive-power signatures.

**Medicine**: initiation of methadone and repeated at 30 days and then annually to evaluate the QT interval as well as if the methadone dose >100 mg/day or if the patient experiences [unexplained] (coefficient = 0.157)

**Humanities**: His considerable political power to her own ends. SALEM's dramatic tension flows from the relationship between Mary Sibley and John Alden, who still love each other but now find themselves [antagonists] (coefficient = 0.153)

**Math**: small shifts in AD, either to the right or the left, will have relatively little effect on the output level Yn, but instead will have a greater effect on the [price] (coefficient = 0.059)

**Coding**: Raven - The central symbol of the story that represents depression and evil. The narrator's name-calling of the bird escalates into a rant by the narrator, including: wretch, thing of [evil] (coefficient = 0.073)

**Logic**: `code "CUN" => 'L', "CCN" => 'P', "CGN" => 'R', "ACN" => 'T', "GUN" => 'V', "GCN" => 'A', "GGN" => 'G', "UCN" => 'S') function string_translate(seq::AbstractString) @assert [length(seq)]` (0.096)

**Instruction following**: services include general printing, variable data printing, security printing as well as document and imaging management solutions. PNMB is located in Putrajaya, a planned city located 25km south of the capital of [Malaysia] (-0.101)

**AbsenceBench** (Fu et al., 2025): It's a very useful screening tool, ACENET, she says, is essential for her work. Some of these models can take days to run without [ACENET] (-0.037)

