# OpenReview forum: "Mapping Overlaps in Benchmarks through Perplexity in the Wild"
_ICLR.cc/2026/Conference — ICLR 2026 Poster_

### Official Review · Reviewer_nELw · 2025-10-25

**Soundness:** 3
**Presentation:** 2
**Contribution:** 2
**Rating:** 4
**Confidence:** 4

**Summary:**

The paper introduces the concept of benchmark signatures that can be used to characterize and compare LLM benchmarks. The signatures are defined as a set of salient tokens drawn from large corpora such that the perplexity of various LMs on those tokens predict their performance on a benchmark. The paper’s contributions are (1) a novel two-stage pipeline to derive “fingerprints” of a benchmark. The pipeline involves perplexity-based screening to identify candidate tokens, and then applying stepwise regression to select a spare set of tokens more predictive of performance. (2) A framework for multi-level benchmark comparison. (3) Empirical insights into how current LLM benchmarks may be redundant or divergent.

**Strengths:**

- The introduction of benchmark signatures is a new way to quantify what a benchmark tests. It introduces a principled multi-level framework for analyzing benchmark redundancy.
- The idea of a misaligned benchmark intent – e.g., logical reasoning benchmarks ending up testing a model’s instruction-following capability – is a notable discovery worthy of further investigation.

**Weaknesses:**

- The paper’s general hypothesis that pretraining familiarity affects performance has been noted in works on benchmark saturation. For example, Humanity’s Last Exam (HLE) was created specifically to address the saturation of MMLU by providing expert-written questions. The paper doesn’t mention HLE or similar broad evaluations that address saturation or overlapping knowledge issues.

      Phan, L., Gatti, A., Han, Z., Li, N., Hu, J., Zhang, H., ... & Wykowski, J. (2025). Humanity's last exam. arXiv preprint arXiv:2501.14249

- It appears the evaluation focuses on academic QA-style tasks (e.g., MMLU, Big-Bench Hard), many of which are multiple-choice or short-answer. This omits classes of benchmarks, such as open-ended generation tasks like summarization or dialogue, where the signature approach may be harder to apply because it inherently works better with well-defined quantitative performance metrics. This focus should be acknowledged as a limitation.

- Some choices in the signature extraction pipeline could use further justification. The authors use a fixed threshold (top/bottom 1% by thrush correlation) for pre-filtering, and then a greedy forward selection up to 1500 tokens. Why 1% and how was 1500 chosen as the max features? There’s no comparison to regularization methods like Lasso or Elastic-Net, which the authors mention as a possible extension but do not report results for.

- A conceptual weakness is the paper’s implicit conflation of model capability and pretraining familiarity. By design, the signature tokens are those that models find easier (lower perplexity) and that correlate with better benchmark scores. This risks a tautology since models may perform well on a benchmark because they have seen similar content before. For benchmarks aiming to test reasoning beyond memorized knowledge, a high signature overlap might indicate the benchmark is inadvertently easier for models that have seen certain clues. In other words, the approach might not distinguish generalization ability from training-set overlap.

**Questions:**

-  If classical regression methods are ill-posed for this high-dimensional problem, did the authors consider regularization techniques or matrix factorization/filtering methods? The use of Lasso or Elastic Net regression is mentioned as an extension but is not explored in this paper.
- For the THRUSHPREFILTER step, the paper mentions using "approximately the top 1% of tokens with the strongest signal” but doesn't specify the exact threshold. Is there a more principled data-driven procedure for selecting this threshold; for example, using cross-validation or analyzing the correlation distribution?
- Relatedly, how sensitive are the results to the fixed variable choices, such as 1% thrush filtering and up to 1500 tokens used in forward selection?
- While code is provided, certain details of the experimental setup are not explicit in the paper. For example, data processing details for extracting signatures from the RedPajama corpus would be helpful; e.g., how the corpus was tokenized, how large $d$ was, and how contexts for tokens were chosen.
- The paper would be stronger if it included concrete examples of signature tokens or token clusters for a few benchmarks. Including a table of signature tokens for different benchmarks and their regression coefficients would enhance interpretability.

---

> ### Author Response · Authors · 2025-11-24
> **Response to Reviewer nELw (Part 1/3)**
>
> We thank the reviewer for the dedicated review. Below is our careful, point-by-point response to your questions & suggestions.
>
> ### **Response to Weakness #1 (Humanity's Last Exam; HLE)**
>
> Thanks for mentioning this important benchmark. Our contribution is orthogonal and complementary: HLE proposes a harder benchmark mainly to combat the **saturation issue** in LLM evaluation, while we explain why existing benchmarks saturate and overlap in the first place. This mechanism applies across MMLU, BBH, and other broader evaluation settings -- regardless of difficulty. We have updated the related-work section to explicitly discuss HLE and related ones to clarify that our work addresses a distinct scientific question: understanding the drivers of benchmark saturation rather than introducing new benchmarks.
>
> ### **Response to Weakness #2 (Other classes of benchmarks)**
>
> Thanks for raising this point and we agree that it is a limitation. While our approach is not conceptually limited to QA/TF tasks, applying it to open-ended generation tasks (e.g., summarization, long-form reasoning, dialogue) requires additional design choices around defining stable, reproducible scoring functions, which itself is far from an easy task. We will clarify our scope in the revised paper.

---

> ### Author Response · Authors · 2025-11-24
> **Response to Reviewer nELw (Part 2/3)**
>
> ### **Response to Weakness #3 (Design choices)**
>
> We agree with your suggestion that certain design choices would benefit from further justification.
>
> **Clarification on feature count and stopping criteria**
>
> There appears to be a slight misunderstanding regarding the feature count. We **do not** set a hard cap of 1500 tokens for the final signature. Instead, the signature size is determined dynamically by the **Akaike Information Criterion (AIC)**. The forward selection process stops **automatically** when adding a new feature no longer improves the information-theoretic trade-off between model fit and complexity (i.e. AIC stops growing). Also as a clarification: our algorithm typically converges at ~30 salient tokens per benchmark (Section A.5, L1071).
>
> **Motivation for the 1% pre-filtering threshold**
>
> The pre-filtering step is guided by both statistical and computational considerations.
>
> **Statistically**, the distribution of robust feature–outcome correlations in our dataset is approximately bell-shaped (see Fig. 3). The 1% threshold (capturing the top/bottom tails) is a conservative heuristic designed to be broad enough to encompass the heavy-tailed components (roughly $> 2.3$ standard deviations in a normal approximation) where signal appears concentrated.
>
> **Computationally**, the subsequent regression step scales as $O(md'^2)$, where $d'$ is the number of features after pre-filtering. Applying a 1% cut reduces dimension by two orders of magnitude, ensuring that the second stage remains tractable.
>
> The overall idea is that any thresholds substantially above 1% make the pipeline computationally difficult, while thresholds too small might risk filtering away important features.
>
> To demonstrate that our chosen preselect ratio is a **robust parameter** rather than an arbitrary choice, we performed a fine-grained sensitivity analysis on the Big-Bench Hard (BBH) dataset. We examined the structural evolution of the model's understanding across a geometric grid of ratios $r \in \{10^{-6}, \dots, 1.0\}$.
>
>
> _Experiment Setup:_
>
> For each ratio in the grid, we generated a $27 \times 27$ inter-benchmark correlation matrix (heat matrix) representing the pairwise relationships between tasks. To quantify the stability of these structures, we compared the heat matrix at ratio $r_i$ to the heat matrix at the next increment $r_{i+1}$. We flattened the upper triangle of each matrix and computed the correlation ($\rho$) between the two steps.
>
> _Results:_
>
> The table below tracks the stability of the heat matrices, where the value in each cell in the right column represents the correlation between the matrices constructed under $r_i$ and $r_{i+1}$.
>
> | Grid Transition ($r_i \to r_{i+1}$) | Heat Matrix Correlation ($\rho$) |
> | :--- | :--- |
> | $10^{-6} \to 10^{-5}$ | $0.2429$ |
> | $10^{-5} \to 10^{-4}$ | $0.3923$ |
> | $10^{-4} \to 10^{-3}$ | $0.6310$ |
> | $10^{-3} \to 0.01$ | $0.9143$ |
> | $0.01 \to 0.1$ | $0.9837$ |
> | $0.1 \to 0.2$ | $0.9990$ |
> | $0.2 \to 0.3$ | $1.0000$ |
> | $0.3 \to 0.4$ | $1.0000$ |
> | $0.4 \to 0.5$ | $1.0000$ |
> | $0.5 \to 1.0$ | $1.0000$ |
>
>
> _Conclusion:_ Visual inspection of the heat matrices combined with the quantitative correlation analysis reveals a clear phase transition. At low ratios ($r < 10^{-3}$), the benchmark relationships are volatile. However, the structure stabilizes significantly by $r=0.01$ ($\rho > 0.91$) and effectively converges by $r=0.1$ ($\rho > 0.98$). Beyond $r=0.2$, the heat matrices become identical ($\rho \approx 1.0$), confirming that selecting a ratio between $0.01$ and $0.1$ efficiently captures the stable, intrinsic structure of the BBH tasks without needing full feature saturation. The benefit of reducing computational complexity far outweighs the cost of sensitivity to parameters.

---

> ### Author Response · Authors · 2025-11-24
> **Response to Reviewer nELw (Part 3/3)**
>
> ### **Response to Weakness #4 (Conceptual tautology)**
>
> Thanks for raising this important conceptual question. We would like to clarify our design logic as follows:
>
> - Our approach is not a tautology. We define signature tokens not by low perplexity, but by their **predictive power** -- the extent to which cross-model perplexity patterns correlate with benchmark outcomes. By computing perplexities on an unrelated "in-the-wild" corpus and regressing them against benchmark scores, we identify tokens whose uncertainty patterns -- either high or low -- track performance rankings. A signature therefore captures the structural alignment between capabilities formed during training and the requirements of the benchmark, not just whether models have "seen" the content (thus only select low-perplexity cases).
> - Our method is designed to study the **coupling** between capacity and familiarity, not to separate them. Perplexity reflects "training exposure + acquired capability" -- a form of capability familiarity. We do not attempt to distinguish direct memorization from statistical learning or architectural bias, because our goal is cross-model comparison: for a given benchmark, on which capability dimensions do high-performing models show consistent familiarity characteristics?
> - High signature overlap is itself *diagnostic*. If a benchmark that claims to test reasoning overlaps heavily with knowledge, that might indicate a design flaw -- and signature overlap ***tells us exactly this***. It reveals what capability dimensions the benchmark actually measures, not what it claims to measure. It tells us when new benchmarks give us new information and when they don’t.
>
> We will revise the 2nd paragraph of Section 4.3 to highlight these points.
>
>
> ### **Response to Question #1**
> Thanks for the question. A small clarification: classical penalized regression methods are **not** "ill-posed" in the sense that a solution does not exist, but directly applying them without the initial correlation screening is computationally intractable. Our extraction pipeline is fully compatible with alternative regularization techniques (e.g., sparse PCA), but within the scope of this paper, we will focus on penalized linear regression. However, we believe that the initial correlation screening stage is still necessary (regardless of the regularization method) to reduce the feature space to a manageable size $d'$ first. In **Point (ii) of general response**, we have conducted ablation studies using Lasso and Elastic Net, which confirm that our extracted signatures and overlap results remain robust across different regularization techniques.
>
> ### **Response to Question #2 and Question #3**
> Please see our response to *Weakness #3* above.
>
> ### **Response to Question #4**
> Thanks for the suggestion. We agree that certain experimental details might not be clear enough in the current form. We will rewrite a more detailed walkthrough of the experiment in the appendix. We will also be open-sourcing our processing codebase/perpelexity and regression results once the double-blind review process is over.
>
> ### **Response to Question #5**
> We conducted a qualitative interpretation of signatures. We find that the knowledge signature is semantically closer to the knowledge of the field, but the signature of general meta-ability benchmarks is not that close, for instance, the detection of missing information, as shown in **Point (iii) of general response**.
>
> ---
>
> Thanks again for taking your valuable time reviewing the paper and giving us feedback. We hope the above responses address your concerns appropriately, and we are open to any further questions and suggestions.

---

> > ### Comment · Reviewer_nELw · 2025-11-25
> >
> > Thank you for the detailed response. I appreciate the authors' commitment to include HLE and related baselines in the final version, which addresses one of my concerns about benchmark saturation. The authors also clarified my concerns around design choices (i.e., stopping criteria and 1% threshold) and how the proposed framework aims to diagnose benchmark overlap rather than isolate generalization. I've updated the overall score accordingly.

---

> > > ### Author Response · Authors · 2025-11-25
> > > **Thank you!**
> > >
> > > Thank you! We’re glad our responses helped clarify your concerns. We appreciate your thoughtful review comments again - they have greatly improved our paper.

---

### Official Review · Reviewer_MMq7 · 2025-10-27

**Soundness:** 3
**Presentation:** 2
**Contribution:** 4
**Rating:** 8
**Confidence:** 4

**Summary:**

This paper introduces benchmark signatures to quantify overlaps among LLM benchmarks by mining salient tokens from in-the-wild corpora where token-level perplexity predicts benchmark performance. Using 32 LLMs and 88 benchmarks, the authors apply Thrush correlation filtering and stepwise forward selection to extract signatures. The work examines three overlap levels: semantic similarity, performance correlation, and signatures. Key findings show signatures discriminate better than semantic or performance measures, performance correlations are biased by format and benchmark family, and coding is the least overlapping domain while logic, math, language, and instruction-following are interconnected.

Despite the writing issues, I highly recognize the motivation of this work. I believe the problem this paper addresses is extremely important and has troubled many researchers working on large language model pretraining. The perspective that in-the-wild corpora implicitly encode the benchmark signature is quite interesting. The discovery that performance-based benchmark agreement is contaminated by format and family effects is valuable for the community. The community should encourage bold papers that tackle important problems. Soundness can be gradually supplemented during the rebuttal phase. This represents ambitious thinking we need for fundamental issues in LLM evaluation. With revisions addressing scalability and adding qualitative analysis of signatures, this will be a strong contribution.

**Strengths:**

The motivation is strong. Benchmark submissions have grown sevenfold and we need methods to understand if new benchmarks measure distinct capabilities or just repackage existing ones. The insight that in-the-wild corpora implicitly encode benchmark signatures through perplexity patterns is interesting and provides a theoretical connection between pretraining exposure and benchmark performance.

The experimental scope with 32 models and 88 benchmarks is substantial. The finding that performance correlations are more influenced by question format and benchmark family than actual function (Figure 1) exposes a real limitation in current benchmark agreement studies. The observation that coding benchmarks are relatively clean and distinct is useful.

The statistical framework using Thrush correlation screening and AIC-based forward selection is reasonable for the ultra-high dimensional setting.

**Weaknesses:**

The writing has issues. It feels like the authors have limited writing experience. The abstract is not very clear and could benefit from referencing how other papers structure theirs. Also, the last line of the last page is not aligned to the bottom. This directly affects the clarity of presentation.

My main concern is computational cost and scalability. The authors use only 1B tokens from RedPajama, then downsample by 50x to ~16.9M token contexts. Our large model in-the-wild datasets are on the scale of tens of trillions of tokens. It is unclear how this work would handle larger scales. From Section A.5.1:
- Initial scale: ~8.45 × 10^9 tokens (RedPajama 1B variant)
- Downsampling: 1/50
- Final scale: ~1.69 × 10^7 tokens
- Feature matrix: P ∈ R^(32 × 1.69×10^7)

The complexity of generalizing this needs discussion. A previous work addressing similar scale is LESS [1]. Getting scaling behavior from 1B → 10B → 100B tokens seems quite difficult, but the authors should discuss this.

The paper never shows what the actual salient tokens are. They extract ~30 tokens per benchmark but provide no examples. What tokens predict math performance? What about coding? Without this, the signatures remain black-box features. This is a significant missed opportunity.

Methodological choices lack justification. The authors mention Lasso or Elastic Net but choose forward selection for "interpretability" without demonstrating this. No comparison with Spearman correlation or mutual information for screening. More ablations would help.

The paper discusses two interpretations of overlap - genuine cognitive interdependence versus "leaky" benchmarks - but never resolves this. When instruction-following and logic overlap more than within-logic benchmarks, the authors could use signature tokens to distinguish these interpretations but do not.

[1] Chen et al., "LESS: Selecting Influential Data for Targeted Instruction Tuning," ICML 2024.

**Questions:**

Can you provide analysis of how signatures change with corpus size? Even experiments with 100M, 500M, and 1B token subsets would show whether signatures stabilize. What is the computational cost and how does it scale to 10B or 100B tokens?

Please show examples of actual salient tokens. What are the top tokens for MMLU math versus MBPP coding? Do they make intuitive sense?

When instruction-following and logic show high signature overlap, what do the shared tokens look like? This could distinguish between genuine interdependence versus contamination.

Have you validated signatures on held-out models not used in extraction? This would show signatures capture fundamental properties rather than overfitting.

---

> ### Author Response · Authors · 2025-11-24
> **Response to Reviewer MMq7 (Part 1/2)**
>
> We very much appreciate the reviewer’s positive feedback.
>
> ### **Response to Weakness #1 (Writing)**:
>
> Thanks for pointing out our writing issues. We will make sure to polish the overall writing, including a rewrite of the abstract and a restructure of the last page, with the goal of increasing presentation clarity. The manuscript will be updated shortly to reflect these changes.
>
> ### **Response to Weakness #2 & Question #1 (Computational Cost & Scalability)**:
>
> Thanks for raising an important question. Our response below is structured in two parts. We will first explain the computational cost, and then use experiments to show why our method is **less vulnerable to the scalability challenge**.
>
> **Computational cost and scalability:**
>
> Admittedly, we face computational constraints similar to LESS [1]. Because Stepwise AIC scales linearly with sample size ($N$), processing larger corpora becomes difficult. At 1B tokens, perplexity extraction takes ~2 hours per model, followed by 30–40 minutes of pre-filtering + regression per benchmark. Scaling to 10B would explode these times to ~20 hours for extraction and ~5 hours per benchmark for regression. This renders 100B+ scales computationally infeasible for our condition.
>
> However, we found that 1B tokens are sufficient for capturing robust structural relationships among the benchmarks. Our experiments below, which vary the amount of tokens from 100K to 10B (10x the original), clearly demonstrate that the stability of our method exhibits strong diminishing returns beyond the 1B-token threshold.
>
>
> **Stabilization Analysis:**
>
> Using a fixed 1% pre-selection ratio, we sampled corpora at logarithmic intervals (100K–10B) and computed task-to-task correlation matrices for 27 BBH sub-tasks under each scale of sampled corpora. Stability was measured via Spearman correlation between consecutive scales:
> + 100K vs 1M: $\rho = 0.278$
> + 1M vs 10M: $\rho = 0.187$
> + 10M vs 100M: $\rho = 0.475$
> + 100M vs 1B: $\rho = 0.998$
> + 1B vs 10B: $\rho = 1.000$
>
>
> **Conclusion on Scalability:**
>
> Results above demonstrate a "phase transition": signatures are unstable at <100M but converge ($\rho > 0.99$) at 100M–1B. Since 1B tokens act as a sufficient statistical proxy, we can use a manageable subset rather than scaling linearly to trillions of tokens. The empirical study further proved that what we need is an appropriate down-sampling, which yields nearly identical results but saves approximately 10 times the resources.
>
> Results above also show that 100M tokens are empirically sufficient to capture signatures without large-scale regression costs. Even though our study employed a sufficient sample size (1B), we emphasize that appropriate downsampling is also feasible for researchers under computational constraints but still want to reproduce our results.
> ___
> [1] Xia, Mengzhou, et al. "Less: Selecting influential data for targeted instruction tuning." ICML 2024.

---

> ### Author Response · Authors · 2025-11-24
> **Response to Reviewer MMq7 (Part 2/2)**
>
> ### **Response to Weaknesses #3, #5 & Questions #2, #3 (Salient Tokens & Signature Overlap)**
>
> We thank the reviewer for their comments that would increase the interpretability of signatures.
>
> What the actual salient tokens are: We conducted a qualitative interpretation of signatures.
> We find that the knowledge signature is semantically closer to the knowledge of the field, but the signature of general meta-ability benchmarks is not that close, for instance, the detection of missing information.
>
> Using shared silent tokens to distinguish between genuine interdependence versus contamination: We want to clarify that benchmark overlap here refers to how similarly models are confused by two sets of silent tokens - not to the semantic or textual overlap of the signature content.
>
> We have a full discussion as shown in the **Point (iii) of general response**.
>
>
> ### **Response to Weakness #4 & Question #4 (Robustness Test)**
> We conducted ablation studies of different regulation methods as shown in the **Point (ii) of general response**. Major conclusions are consistent.
>
> **Spearman correlation or mutual information for screening:**
>
> We thank the reviewer for suggesting alternative selection methods in the preselection phase. To validate our choice, we compared Thrush against Spearman correlation and Mutual Information (MI). We normalized all scores to a [0, 1] scale and analyzed the standard deviation (Std) of their distributions across benchmarks.
> Our analysis reveals that Mutual Information consistently exhibits a lower Std, indicating significantly lower discriminative power compared to the ranking metrics.
>
> | Benchmark       | Metric       | Std (0-1 Scale) |
> | :---       | :---         | ---:            |
> | MBPP       | thrush       | 0.1549          |
> | MBPP       | spearman     | 0.1562          |
> | MBPP       | mutual_info  | 0.1057          |
> | IFEVAL     | thrush       | 0.1557          |
> | IFEVAL     | spearman     | 0.1557          |
> | IFEVAL     | mutual_info  | 0.0926          |
> | BBH        | thrush       | 0.1808          |
> | BBH        | spearman     | 0.1808          |
> | BBH        | mutual_info  | 0.0981          |
>
> Furthermore, when comparing Thrush and Spearman, we found they produce nearly identical selections, with a 99.5% overlap in selected features. Given this equivalence and its high discriminative capability, we maintained the use of Thrush.

---

> > ### Comment · Reviewer_MMq7 · 2025-11-25
> > **I was indeed not mistaken about this work. 8 score is fair.**
> >
> > I was indeed not mistaken about this work. 8 score is fair.

---

> > > ### Author Response · Authors · 2025-11-25
> > > **Thank you!**
> > >
> > > Thank you! We appreciate your thoughtful review comments again - they have greatly improved our paper.

---

### Official Review · Reviewer_tBme · 2025-11-01

**Soundness:** 4
**Presentation:** 3
**Contribution:** 3
**Rating:** 6
**Confidence:** 4

**Summary:**

This paper introduces benchmark signatures: sets of salient tokens from in-the-wild corpora where LLM perplexity predicts benchmark performance. Using 32 LLMs and 88 benchmarks, the authors compare three overlap measurement approaches: semantic (question similarity), performance (score correlations), and signature-based (perplexity patterns). Key findings: (1) signature analysis better distinguishes benchmarks than semantic/performance approaches, (2) performance-level results are strongly influenced by benchmark-orthogonal factors, (3) coding is the most distinct domain while math/logic/language show cross-functional overlap.

**Strengths:**

1. I like the idea of signature. The signature approach elegantly unifies semantic and performance perspectives: signatures inherently contain semantic information (salient tokens from meaningful contexts) while perplexity directly hints at performance. This is a creative and well-motivated synthesis.
2. The experiments are done in Impressive scale: 32 models × 88 benchmarks with multi-level analysis provides comprehensive coverage.
3. There are good theory grounding. Good justification via SIS theory for ultra-high dimensional feature selection and two-stage filtering approach appropriately handles the d >> m problem.

**Weaknesses:**

1. All signatures are extracted from RedPajama only, yet the 32 models were trained on diverse corpora. It remains unclear whether RedPajama signatures are representative of these varied training distributions. Furthermore, proprietary models may rely on private training data containing unconsidered signatures. If signatures are biased, can they still meaningfully reflect model capabilities? Robustness experiments are needed to validate the approach.
2. The claim that signature correlations reveal "model capabilities" is questionable. Two alternative explanations exist: (a) High within-dataset vs. between-dataset correlations may simply reflect training data contamination, as authors acknowledge; (b) Same dataset may inherently represent a single capability independent of intended subject divisions—for instance, if all MMLU subjects source from Wikipedia, signatures may capture "Wikipedia-style capability" (or even spurious correlation) rather than genuine subject-specific skills (history vs. chemistry). The perplexity-performance link could be coincidental. Can you do more causality analysis on that?

**Questions:**

See weakness.

---

> ### Author Response · Authors · 2025-11-24
> **Response to Reviewer tBme (Part 1/2)**
>
> We very much appreciate the reviewer’s relatively positive feedback and insightful comments on data representativeness and the general ability effect.
> ### **Response to Weakness #1**:
> - While it is true that our signatures are extracted from RedPajama, this does not undermine their relevance. RedPajama is broadly representative of “in-the-wild” web data including Wikipedia, GitHub, C4, arxiv, etc, which forms the dominant component of modern LLM pretraining. As noted in [1], LLMs are increasingly converging because major model families are trained on similar mixtures of large-scale web corpora, code, and curated text. In other words, although individual datasets differ at the margins, they share substantial structural and statistical overlap. Crucially, our goal is not to reconstruct or identify the exact training data of any model. Rather, we aim to capture generalizable distributional signatures that emerge across large in-the-wild corpora. Because RedPajama reflects the broad characteristics of public web text -- and because model training corpora largely draw from the same underlying data universe -- RedPajama provides a sufficiently representative substrate for extracting robust signatures. Thus, the method does not depend on exact training data matching. Instead, it leverages the empirical regularities of large-scale in-the-wild text, which are shared across most contemporary LLMs.
> - Second, to validate the robustness of our signatures against training data variations, we conducted a **control experiment using Dolma** [2], another massive training dataset derived from diverse sources (including C4, arXiv, and others). We replicated our exact pipeline on Dolma: preprocessing, downsampling, and extracting perplexities to generate signatures for all BBH subtasks and computing the correlation matrix that captures the inter-correlation between subtasks. By flattening the upper triangles of the matrices and calculating the Spearman correlation between the matrices derived from RedPajama and Dolma, we obtained a high agreement of $\mathbf{\rho = 0.895}$. This strong correlation demonstrates that the task signatures are robust to the specific choice of corpus, provided that the data is a sufficiently large and representative sample of in-the-wild data.
>
>
> ---
> [1] Wolfram, Christopher and Aaron Schein. "Layers at Similar Depths Generate Similar Activations Across LLM Architectures." COLM 2025.
>
> [2] Soldaini, Luca, et al. "Dolma: An open corpus of three trillion tokens for language model pretraining research." ACL 2024.

---

> ### Author Response · Authors · 2025-11-24
> **Response to Reviewer tBme (Part 2/2)**
>
> ### **Response to Weakness #2**:
> We thank the reviewer for this insightful observation. To empirically adjudicate whether our signatures capture distinct, subject-specific factors versus a generic "dataset style" (e.g., a common "factor" shared across all BBH/Wikipedia tasks), we conducted a Cross-Task Transfer Experiment. If the alternative hypothesis is correct -- that signatures primarily capture a generic capability or dataset artifact -- a signature derived from Benchmark A ($S_A$) should successfully predict the performance on a different Benchmark B ($Y_B$), provided they share that generic source.
>
> _Experiment Setup:_
>
> We performed the following for all 27 Big-Bench Hard (BBH) subtasks:
> For a given benchmark $A$, we fixed its covariate feature matrix from its signatures $X_A$. We then trained 26 separate regression models, each regressing the performance vector of a different benchmark $Y_B$ (where $B \neq A$) against $X_A$ and collecting the absolute error of the prediction on B from $X_A$ against the true value of $Y_B$. We iterated this process for all $A$ (all subtasks in BBH), resulting in a comprehensive evaluation of how well one task's signature predicts another task's performance. We also refit $X_B$ vs $Y_B$ again and extract the absolute error each time to obtain the task-specific baseline.
>
> _Results:_
> + Baseline (predict based on mean; LOOCV from the general response): $0.181 \pm 0.144$
> + Cross-Task Prediction: $0.223 \pm 0.048$
> + Task-specific baseline: $0.021 \pm 0.012$
>
>
> The observed MAE is significantly larger when attempting to use one task's signature to predict another compared to using the task's own signature ($0.223$ vs $0.021$). Crucially, the cross-task error is even higher than the naive LOOCV baseline ($0.181$), where we simply used the mean of the performance vector. If the signatures **merely** encoded a generic "style" or shared data contamination, the covariates of Task A ($X_A$) would serve as a sufficient proxy for the capabilities required by Task B ($Y_B$). Under this hypothesis, training on $X_A$ to predict $Y_B$ should yield a good fit. The observed degradation in performance contradicts this, indicating that $X_A$ encodes specific, non-transferable structural information unique to Task A.

---

> ### Comment · Reviewer_tBme · 2025-11-25
> **Official Comment by Reviewer tBme**
>
> Thank you for your detailed response. I think the paper is acceptable, most of my concerns have been addressed, and I will raise my score.

---

> > ### Author Response · Authors · 2025-11-25
> > **Thank you!**
> >
> > Thank you! We’re glad our responses helped clarify your concerns. We appreciate your thoughtful review comments again - they have greatly improved our paper.

---

### Official Review · Reviewer_XQJb · 2025-11-04

**Soundness:** 3
**Presentation:** 3
**Contribution:** 4
**Rating:** 8
**Confidence:** 4

**Summary:**

This paper exposes a critical flaw in benchmark categorization that benchmark overlap is dominated by superficial factors like quention format and benchmark family, rather than the actual capabilities, and proposes "Benchmark signatures" to reliably describe benchmark features. Key findings from this analysis include:
- Logic, math, IF and language form an interconnected cluster
- Coding stands distinctly alone from other domains
- Many "logic" benchmarks actually measure instruction-following
- Multilingual / cultural benchmarks show low within-cateogry similarity
The takeaway is the current evaluation methods conflate test design with capability. Signature-based analysis reveals what LLMs actually learn.

**Strengths:**

1. The benchmark signature concept is novel and elegantly bridges pretraining data characteristics and benchmark performance through perplexity. The mechanistic view (rather than correlational view) reveals intrinsic capabilities that benchmarks measure rather than superficial artifacts.
2. The paper repurposes proven perplexity-correlation methods from data selection literature for benchmark characterization, and provide rigorous theoretical justification through SIS framework. It explains why the proposed method work through rigorous theoretic analysis.
3. The paper reveals that cross-function overlap (instruction following vs. logic) exceeds within-function overlap in some cases, shows interesting observation that some benchmarks designed to test "logic" acutually measure instruction following, and this further demonstrates the power of the methodology which leads to non trivial findings.

**Weaknesses:**

1. Table 1 shows token level has the greatest standard deviation and interquartile range, and the paper claims this indidcates token level is the most informative. However, this is not necessarily true: the higher vairance could also originates from larger noise. It is better to show signal-to-noise ratio analysis (like R^2) for token vs. chunk vs. doc level.
2. All analysis uses the same 32 models for extraction and comparison. Cross-validation and held-out test results are absent.
3. Across 16 targets, token-level values achieved 12 wins, which indicates 25% of loses. Please provide analysis of when and why token level loses.
4. AIC is a greedy algorithm known to be suboptimal, why not use LASSO, elastic net, etc.? Could you provide stability analysis of feature selection?

**Questions:**

1. Can signatures from 24 models predict the performance ranking of the held-out 8 models on benchmarks (held-out test for predictive validation)?
2. What are the examples of actual salient tokens for specific benchmarks? Do they align with human intuition?
3. Please add confidence intervals to Fig. 4 and conduct pairwise tests for within category vs. cross category differences.

---

> ### Author Response · Authors · 2025-11-24
> **Response to Reviewer XQJb**
>
> We thank the reviewer for their careful review and for acknowledging our contributions. Below we will carefully respond to your questions & suggestions.
>
> ### **Response to Weakness #1**
>
> We agree that $R^2$ is a very informative metric to distinguish signal from noise. In Table 1 below (copied from revised Appendix A.2), we include the Adjusted Coefficient of Determination ($\mathbf{R^2_{adj}}$), derived from the actual fit of the linear model following the regression setup discussed in the main paper. We report $\mathbf{R^2_{adj}}$ rather than standard $R^2$ to account for the degrees of freedom relative to our sample size, ensuring that higher scores reflect genuine predictive signals rather than overfitting. Across the 4 benchmarks, the token level achieves the highest $\mathbf{R^2_{adj}}$ scores 3 times (sometimes as high as 0.9), while the document level consistently results in the lowest performance. This empirical evidence directly addresses the concern regarding variance versus noise: if the higher variance observed at the token level were merely noise, it would result in a poor model fit.
>
> | **Benchmark** | **Level** | **Std** | **IQR** | **Max–Q99** | **Q01–Min** | $\mathbf{R^2_{adj}}$ |
> |:--------------:|:---------:|--------:|--------:|------------:|------------:|------------:|
> | Gsm8k | Chunk | 30.30 | 44.63 | **43.37** | **36.63** | 0.903 |
> | Gsm8k | Doc | 19.39 | 26.50 | 23.25 | 18.75 | 0.826 |
> | Gsm8k | Token | **36.53** | **54.11** | 40.63 | 27.37 | **0.927** |
> | Mbpp | Chunk | 29.83 | 43.05 | **87.62** | 27.05 | 0.849 |
> | Mbpp | Doc | 14.49 | 18.78 | 32.44 | 8.67 | 0.667 |
> | Mbpp | Token | **36.76** | **50.60** | 39.60 | **39.20** | **0.885** |
> | Mmlu | Chunk | 29.31 | 41.18 | **64.00** | 30.36 | **0.551** |
> | Mmlu | Doc | 18.11 | 27.68 | 35.16 | 9.68 | 0.185 |
> | Mmlu | Token | **38.05** | **51.90** | 52.00 | **42.29** | 0.454 |
> | Truthfulqa | Chunk | 30.58 | 41.80 | 48.00 | 29.20 | 0.414 |
> | Truthfulqa | Doc | 17.67 | 22.75 | 25.00 | 13.00 | 0.208 |
> | Truthfulqa | Token | **36.57** | **53.00** | **49.20** | **35.40** | **0.505** |
>
> _Table 1: updated from Appendix A.2 by adding adjusted $R^2$ metric._
>
> ### **Response to Weakness #2, Question #1, Question #2**
>
> We further conducted held-out and cross-validation experiments, and added concrete examples and interpretations of signatures. Please see **Points (i) and (iii) in our general response**.
>
> ### **Response to Weakness #3**
>
> As shown in Table 1, the token-level measures consistently provide the strongest signal for Std and IQR. While chunk-level occasionally yields larger gaps between extreme values and the top/bottom 1% quantiles. Please note that these tail metrics are sensitive to outliers, which are often noisy and noninformative. Consequently, we mainly rely on IQR and Std for selection. Token-level measures introduce the least inductive bias (minimal structural assumptions and segmentation artifacts) while offering the highest granularity and more faithful representation of model uncertainty. Meanwhile, the updated results in Appendix A.2 further reinforce this decision.
>
> ### **Response to Weakness #4**
>
> Thank you for the suggestion. Please refer to **Point (ii) in general response** for the newly-added ablation studies on regression methods.
>
> ### **Response to Question #3**
>
> We will add the full Appendix A.6 section for the statistical analysis. In short, conclusions hold. First we want to clarify that the confidence interval is not that meaningful in this case due to the fact that some categories have just a few benchmarks. We did the following statistical pairwise analysis: we used a bootstrapping approach to evaluate whether the signature overlaps within a benchmark category differ statistically from those across categories. For each pair of benchmarks, we computed the overlap between their signatures, which reflects how similarly the chosen set of representative LLMs are “confused” by the two benchmarks. Because the numbers of within-category and cross-category pairs differ, we performed 10,000 bootstrap samples to estimate the distributions and corresponding p-values. The resulting difference, expressed as a positive or negative percentile value, indicates how much larger or smaller the within-category mean correlations are compared to the mean cross-category correlation, and whether this deviation is statistically significant.
>
> | Benchmark Category       | Diff  | P-value |
> |--------------------------|-------|---------|
> | humanities               | -46%  | 0.00    |
> | reasoning                | +21%  | 0.00    |
> | language                 | -10%  | 0.15    |
> | medicine                 | +6%   | 0.54    |
> | science and engineering  | +42%  | 0.00    |
> | social science           | +41%  | 0.00    |
> | world knowledge          | -17%  | 0.01    |
> | business              | +2%  | 0.70    |

---

### Author Response · Authors · 2025-11-24
**General response 2**

### **(iii): Qualitatively Interpreting Signatures**

We conducted a qualitative analysis of the textual signatures. Using a simple metric—cosine similarity between a benchmark’s intended function (e.g., social-science knowledge) and the textual content of its signature via MPNet embeddings `MPNet` [1] -- we assessed how well signatures align with benchmark semantics. Highlighted examples see the second response.


When a benchmark targets knowledge in a specific field, its signature tends to reflect that semantic content - the signature is, in effect, “about” the same knowledge domain. In some cases, the cosine similarity reaches as high as 0.4. On the other hand, some benchmark signatures bear little relation to their intended functions, such as logical reasoning and `AbsenceBench` [4] (a benchmark designed to measure an LLM’s ability to identify missing information in a sequence (essentially a reverse Needle in a Haystack task)).


#### **Why do some benchmark signatures “match” the stated function while others don’t?**

- Benchmarks often bundle multiple subskills: beyond the target ability, they depend on instruction following, reading load, and format handling. As a result, signature tokens often reflect whichever auxiliary factor drives the most performance variance. Knowledge benchmarks are cleaner, while abstract meta-ability tasks (e.g., logical reasoning, detecting missing information) are more distorted by these side demands and by gaps between task design and implementation. For example, for Absencebench, no signatures have a super high predictive power (the highest 0.03).
- Signatures come from predictive token-level perplexity in natural corpora; when the intended skill is rare or procedural (like “logical reasoning”, “detect missing information”), models default to proxy cues—genre, discourse markers, instruction tokens—rather than domain-specific features. This problem is smaller for well-defined knowledge areas. Signatures often include numerals, syntax tokens, or discourse markers that look semantically unrelated, whereas knowledge tasks appear more semantically aligned simply because their signatures form coherent domain narratives.
- Finally, strong predictive power doesn’t imply shared semantics: models can rely on statistical co-occurrences in the natural corpora correlated with appearances of benchmark questions rather than true semantic relations. Semantic embeddings therefore cannot fully approximate models’ internal task representations, consistent with findings of transferable but non-human-interpretable structures [2, 3]. Benchmark overlap here refers to how similarly models are confused by two sets of silent tokens - not to the semantic or textual overlap of the signature content.

We show benchmark example signatures and coefficients here. The last part is the silent token with the coefficient in parentheses. They are all high-predictive-power signatures.

- **Medicine**: *initiation of methadone and repeated at 30 days and then annually to evaluate the QT interval as well as if the methadone dose >100 mg/day or if the patient experiences **unexplained*** (0.157)
- **Humanities**: *His considerable political power to her own ends. SALEM’s dramatic tension flows from the relationship between Mary Sibley and John Alden, who still love each other but now find themselves **antagonists*** (0.153)
- **Math**: *small shifts in AD, either to the right or the left, will have relatively little effect on the output level Yn, but instead will have a greater effect on the **price*** (0.059)
- **Coding**: *Raven - The central symbol of the story that represents depression and evil. The narrator's name-calling of the bird escalates into a rant by the narrator, including: wretch, thing of **evil*** (0.073)
- **Logic**: *code "CUN" => 'L', "CCN" => 'P', "CGN" => 'R', "ACN" => 'T', "GUN" => 'V', "GCN" => 'A', "GGN" => 'G', "UCN" => 'S' ) function string_translate(seq::AbstractString) @assert length **(seq)*** (0.096)
- **Instruction following**: *services include general printing, variable data printing, security printing as well as document and imaging management solutions. PNMB is located in Putrajaya, a planned city located 25km south of the capital of **Malaysia*** (-0.101)
- **AbsenceBench** [4]: *It’s a very useful screening tool, ACENET, she says, is essential for her work. Some of these models can take days to run without **ACENET*** (-0.037)

---
[1] Song, Kaitao, et al. "MPNet: Masked and Permuted Pre-training for Language Understanding." NeurIPS 2020.

[2] Wu, Zhaofeng, et al. "The semantic hub hypothesis: Language models share semantic representations across languages and modalities." arXiv preprint arXiv:2411.04986 (2024).

[3] Musker, Sam, et al. "LLMs as Models for Analogical Reasoning." arXiv preprint arxiv:2406.13803 (2025).

[4] Fu, Harvey Yiyun, et al. "AbsenceBench: Language Models Can't Tell What's Missing." NeurIPS 2025.

---

### Author Response · Authors · 2025-11-24
**General response 1**

We thank all reviewers for their valuable comments. Below are some general responses to the common concerns raised regarding **held-out experiments, ablation studies of regularization methods, and concrete examples of signatures**. We are currently polishing the paper and incorporating the new results. *We will upload the revised version shortly*.

### **(i): Leave-One-Out Cross-Validation (LOOCV)**

To assess generalization, we performed LOOCV over 32 models on the 27 BBH sub-tasks, comparing our predictor to a baseline that uses the mean to predict the held-out model's performance. Our model achieved an MAE of $0.076±0.038$, substantially better than the baseline’s $0.181±0.144$. This confirms that the extracted features capture genuine, reusable structure rather than overfitting. We further tested robustness with a strict out-of-sample evaluation using two fully held-out models -- `Qwen2.5-7B` and `Falcon-rw-7b`. Using the 27 training models to derive features and forecast performance across 27 benchmarks, we obtained MAEs of $0.0681±0.034$ and $0.0722±0.026$, respectively. These results show that the model generalizes well even to unseen observations.

### **(ii): Ablation Study of Regularization Methods**
We ran an ablation study replacing AIC-based forward selection (**abbreviated as "AIC" below**) with Lasso ($\alpha=1$) and Elastic Net ($\alpha=1, l1\_{ratio}=0.5$), keeping all other regression settings fixed. For each MMLU benchmark, we derived a signature vector under each method and computed the corresponding inter-benchmark correlation matrix. We then flattened each matrix’s upper triangle and measured **Spearman correlations ($ρ$)** among regression methods to assess whether the structural relationships among benchmarks persist under different regularizers. Results are:

- AIC vs. Lasso $ρ = 0.763$,
- AIC vs. Elastic Net $ρ = 0.765$,
- Lasso vs. Elastic Net $ρ = 0.786$.

As a baseline, using 50 randomly sampled features (after the correlation filtering) produced $ρ = 0.334$ against AIC. These give us two interesting insights:

1. As expected, the initial filtering helps retain (marginally) informative tokens, so signatures constructed from random sampling have a non-trivial but weak correlation with regression-based counterparts.
2. The moderate-to-strong correlations among regression methods show that while the chosen features may vary, **the between-benchmark structural relationships revealed by the signatures remain largely stable across regularization strategies**.

---

### Author Response · Authors · 2025-11-26
**updated abstract**

We sincerely thank the reviewers for their insightful comments, which have greatly improved our paper. As we promised, we've updated the abstract and clarified several technical details and walkthroughs in the main paper, as these were specifically noted as unclear by reviewers. We leave some incomplete parts, which have been explicitly discussed in the rebuttal below. We hope these revisions adequately address the concerns. We also commit to completing the placeholder sections in the main paper to include all results presented in our rebuttal for the camera-ready version.

---

### Author Response · Authors · 2025-12-02
**Final message to AC (Summary of rebuttals)**

Following ICLR’s updated review policy, we sincerely thank you for taking over as our new AC. This comment summarizes our extensive discussions with the four original reviewers and outlines the specific updates made to the manuscript.

### **Consensus of Acceptance**
We are pleased to report a **consensus of strong acceptance** among all reviewers following the rebuttal. As evidenced by the discussion timestamps (Nov 25), all discussions had already concluded fairly positively with satisfaction of our rebuttals before the OpenReview platform issue became public on November 27. **Reviewers tBme (initial rating 6) and nELw (initial rating 4) explicitly stated in their final comments that they have raised their scores on Nov 25.** MMq7 explicitly stated “8 is fair” on Nov 25.

### Response to weakness in Reviewer XQJb’s comments (initial 8 - no response)

*   **Signal vs. Noise:** We addressed concerns regarding token-level variance by calculating the Adjusted Coefficient of Determination ($R^2_{adj}$), proving that token-level features provide the strongest predictive signal rather than noise.
*   **Generalization:** We added Leave-One-Out Cross-Validation (LOOCV) and held-out experiments, achieving a low Mean Absolute Error ($0.07$ vs the baseline using the mean $0.18$), confirming the model generalizes well to unseen observations.
*   **Ablation:** We compared our AIC-based selection against Lasso and Elastic Net, finding strong structural correlations ($\rho > 0.76$), demonstrating method robustness.

### Response to weakness in Reviewer tBme’s comments (initial 6, raised)

*   **Data Bias:** To prove our signatures aren't artifacts of `RedPajama`, we replicated the pipeline on the `Dolma` dataset. The resulting correlations showed high agreement ($\rho = 0.895$), demonstrating the robustness of data selection.
*   **Specificity vs. Generic Capability:** We conducted a Cross-Task Transfer experiment. Results showed that a benchmark’s signature predicts its own performance significantly better than it predicts others, confirming that signatures capture specific structural information rather than generic "dataset style" or contamination.

### Response to weakness in Reviewer MMq7’s comments (initial 8, kept)

*   **Scalability:** We addressed concerns about computational costs by analyzing corpus size scaling. We identified a "phase transition" where signatures stabilize between 100M–1B tokens, confirming our 1B-token subset is statistically sufficient and that the method is computationally feasible under limited computational resources.
*   **Qualitative Analysis:** We provided concrete examples of salient tokens and our interpretations (e.g., knowledge benchmarks align semantically with field knowledge, while general-ability benchmarks often rely on proxy cues), directly addressing the request for interpretability.
* **Screening Methods:** We compared Thrush against Spearman correlation and Mutual Information, and revealed that Thrush shows a high discrimination power.

### Response to weakness in Reviewer nELw’s comments (initial 4, raised)

*   **Clarification of Factual Misunderstanding:** We corrected the misunderstanding that we used a "hard cap" of 1500 tokens; we clarified that feature count is dynamically selected via Akaike Information Criterion.
*   **Tautology Concern:** We clarified that high predictive power is **not** a tautology but a diagnostic tool. We chose signatures according to predictive power (no matter if the perplexity is high or low). We explained that we measure the *coupling* between capacity and familiarity to diagnose benchmark validity (e.g., detecting when a "logic" benchmark actually measures instruction following)
*   **Sensitivity Analysis:** We performed a grid search on the pre-filtering threshold, demonstrating that results are *stable* across a wide range of parameters.



### Updates to the Manuscript
We have uploaded a revised PDF with changes highlighted in red placeholders.

**1. Main Paper:**
*   **New Section 5:** Added qualitative interpretation of signatures to explain *what* the model is actually learning (General Response iii).
*   **Discussion & Future Work:** Expanded to include "Humanity's Last Exam" (HLE) as related work, revised future work to add more types of benchmarks, discussed the distinction between benchmark saturation and our diagnostic analysis (addressing nELw).
*   **Figure 4** and its accompanying narrative have been updated to incorporate statistical significance.


**2. Appendix (Major Additions):**
*   **Robustness Checks (Section A.7):** A comprehensive breakdown of robustness across four dimensions: Design (Generalizability), Methods (Regularization and screening comparisons), Parameters (Threshold sensitivity), and Data (RedPajama vs. Dolma).
*   **Computational Cost (Section A.8):** Detailed analysis of scalability and resource requirements.
*   **Statistical Analysis:** Expanded Table 1 to justify token-level granularity.

---

### Meta-Review · Area_Chair_8BEW · 2026-01-07

**Summary:**

This paper introduces benchmark signatures, sets of salient tokens from in-the-wild corpora whose cross-model perplexity patterns predict benchmark performance. The authors uses them to analyze overlap among 89 benchmarks across 32 LLMs. Reviewers agree that the problem is timely and that the approach offers a novel lens on benchmark overlap and saturation beyond surface-level semantic similarity.

The main concerns focus on validation and robustness (e.g., held-out models and cross-validation), methodological choices (feature selection, thresholds), representativeness of the corpus used for signature extraction, interpretability of the signatures, and conceptual clarity around capability versus pretraining familiarity. The rebuttal substantially strengthens the paper by adding predictive validation, robustness checks across corpora, regression ablations, and scaling analyses, and by clarifying the intended interpretation of signatures. Some issues remain only partially resolved, particularly around interpretability and causal framing, but the core contributions are now better supported.

Please address the following in the revised version: the abstract is unusually long and should be tightened to improve clarity and readability, and several figures would benefit from careful revision to enhance legibility.

Overall, I recommend acceptance. The core idea is interesting and, following the rebuttal, empirically well supported. While the paper would benefit from improved interpretability, additional proofreading, and clearer conceptual positioning, these issues are largely presentational and do not detract from the paper’s main contributions.

**Reviewer Concerns:**

I believe that most reviewers' core concerns are addressed by the rebuttal. Specifically,

Reviewer XQJb's concerns about variance vs. signal, lack of held-out validation, and feature-selection stability are largely addressed through added adjusted R-square, cross-validation, and regression ablations.

Reviewer tBme's concerns about corpus bias and whether signatures reflect true capabilities versus dataset artifacts are meaningfully addressed via robustness experiments and cross-task transfer analyses. Deeper causal interpretation remains a limitation but is now more clearly framed.

Reviewer MMq7 raises issues around scalability and interpretability, which are partially addressed with stabilization and cost analyses. However, clearer presentation of concrete signature tokens in the main text and improved writing remain important.

Reviewer nELw's concerns about related work, scope limitations, and sensitivity of design choices are largely addressed through added discussion and sensitivity analyses.

**Reviewer Scores:**

With full discussion, I would expect modest positive score adjustments across reviewers, reflecting the strengthened empirical validation and clearer positioning after rebuttal. While some higher-level concerns remain, they do not outweigh the paper’s contributions, leading to an overall accept.

---

### Decision · Program_Chairs · 2026-01-26

Accept (Poster)